# Run Away From your Teacher: a New Self-Supervised Approach Solving the Puzzle of BYOL

## Abstract

Recently, a newly proposed self-supervised framework Bootstrap Your Own Latent (BYOL) seriously challenges the necessity of negative samples in contrastive learning frameworks. BYOL works like a charm despite the fact that it discards the negative samples completely and there is no measure to prevent collapse in its training objective. In this paper, we suggest understanding BYOL from the view of our newly proposed interpretable self-supervised learning framework, Run Away From your Teacher (RAFT). RAFT optimizes two objectives at the same time: (i) aligning two views of the same data to similar representations and (ii) running away from the model's Mean Teacher (MT, the exponential moving average of the history models) instead of BYOL's running towards it. The second term of RAFT explicitly prevents the representation collapse and thus makes RAFT a more conceptually reliable framework. We provide basic benchmarks of RAFT on CIFAR10 to validate the effectiveness of our method. Furthermore, we prove that BYOL is equivalent to RAFT under certain conditions, providing solid reasoning for BYOL's counter-intuitive success.

## 1 Introduction

Recently the performance gap between self-supervised learning and supervised learning has been narrowed thanks to the development of contrastive learning (Chen et al., 2020b;a; Tian et al., 2019; Chen et al., 2020b; Sohn, 2016; Zhuang et al., 2019; He et al., 2020; Oord et al., 2018; Hadsell et al., 2006). Contrastive learning distinguishes positive pairs of data from the negative. It has been shown that when the representation space is $l_2$-normalized, i.e. a hypersphere, optimizing the contrastive loss is approximately equivalent to optimizing the *alignment* of positive pairs and the *uniformity* of the representation distribution at the same time (Wang & Isola, 2020). This equivalence conforms to our intuitive understanding. One can easily imagine a failed method when we only optimize either of the properties: aligning the positive pairs without uniformity constraint causes representation collapse, mapping different data all to the same point; scattering the data uniformly in the representation space without aligning similar ones yields no more meaningful representation than random.

The proposal of Bootstrap Your Own Latent (BYOL) fiercely challenges our consensus that negative samples are necessary to contrastive methods (Grill et al., 2020). BYOL trains the model (*online* network) to predict its Mean Teacher (moving average of the online, refer to Appendix B.2) on two augmented views of the same data (Tarvainen & Valpola, 2017). There is no explicit constraint on uniformity in BYOL, while the expected collapse never happens, what's more, it reaches the SOTA performance on the downstream tasks. Although BYOL has been empirically proven to be an effective self-supervised learning approach, the mechanism that keeps it from collapse remains unrevealed. Without disclosing this mystery, it would be disturbing for us to adapt BYOL to other problems, let alone further improve it. Therefore solving the puzzle of BYOL is an urgent task.

In this paper, we explain how BYOL works through another interpretable learning framework which leverages the MT in the exact opposite way. Based on a series of theoretical derivation and empirical approximation, we build a new self-supervised learning framework, Run Away From your Teacher (RAFT), which optimizes two objectives at the same time: (i) minimize the representation

distance between two samples from a positive pair and (ii) maximize the representation distance between the online and its MT. The second objective of RAFT incorporates the MT in a way exactly opposite to BYOL, and it explicitly prevents the representation collapse by encouraging the online to be different from its history (Figure 2a). Moreover, we empirically show that the second objective of RAFT is a more effective and consistent regularizer for the first objective, which makes RAFT more favorable than BYOL. Finally, we solve the puzzle of BYOL by theoretically proving that BYOL is a special form of RAFT when certain conditions and approximation hold. This proof explains why collapse does not happen in BYOL, and also makes the performance of BYOL' an approximate guarantee of the effectiveness of RAFT.

The main body of the paper is organized in the same order of how we explore the properties of BYOL and establish RAFT based on them (refer to Appendix A for more details). In section 3, we investigate the phenomenon that BYOL fails to work when the predictor is removed. In section 4, we establish two meaningful objectives out of BYOL by upper bounding. Based on that, we propose RAFT due to its stronger regularization effect and its accordance with our knowledge. In section 5, we prove that, as a representation learning framework, BYOL is a special form of RAFT under certain achievable conditions.

In summary, our contributions are listed as follows:

- We present a new self-supervised learning framework RAFT that minimizes the alignment and maximizes the distance between the online network and its MT. The motivation of RAFT conforms to our understanding of balancing alignment and uniformity of the representation space, and thus could be easily extended and adapted to future problems.

- We equate two seemingly opposite ways of incorporating MT in contrastive methods under certain conditions. By doing so, we unravel the puzzle of how BYOL avoids representation collapse.

## 2 BACKGROUND AND RELATED WORK

### 2.1 TWO METRICS OPTIMIZED IN CONTRASTIVE LEARNING

Optimizing contrastive learning objective has been empirically proven to have positive correlations with the downstream task performance (Chen et al., 2020b;a; Tian et al., 2019; Chen et al., 2020b; Sohn, 2016; Zhuang et al., 2019; He et al., 2020; Oord et al., 2018). Wang & Isola (2020) puts the contrastive learning under the context of hypersphere and formally showcases that optimizing the contrastive loss (for preliminary of contrastive learning, refer to Appendix B.1) is equivalent to optimizing two metrics of the encoder network when the size of negative samples $K$ is sufficiently large: the alignment of the two augmented views of the same data and the uniformity of the representation population. We introduce the alignment objective and uniformity objective as follows.

**Definition 2.1 (Alignment loss)** *The **alignment loss** $\mathcal{L}_{\text{align}}(f, \mathcal{P}_{\text{pos}})$ of the function $f$ over positive-pair distribution $\mathcal{P}_{\text{pos}}$ is defined as:*

$$\mathcal{L}_{\text{align}}(f; \mathcal{P}_{\text{pos}}) \triangleq \mathbb{E}_{(x_1, x_2) \sim \mathcal{P}_{pos}} \left[ \|f(x_1) - f(x_2)\|_2^2 \right], \tag{1}$$

where the positive pair $(x_1, x_2)$ are two augmented views of the same input data $x \sim \mathcal{X}$, i.e. $(x_1, x_2) = (t_1(x), t_2(x))$ and $t_1 \sim \mathcal{T}_1, t_2 \sim \mathcal{T}_2$ are two augmentations. For the sake of simplicity, we omit $\mathcal{P}_{\text{pos}}$ and use $\mathcal{L}_{\text{align}}(f)$ in the following content.

**Definition 2.2 (Uniformity loss)** *The **loss of uniformity** $\mathcal{L}_{\text{uniform}}(f; \mathcal{X})$ of the encoder function $f$ over data distribution $\mathcal{X}$ is defined as*

$$\mathcal{L}_{\text{uniform}}(f; \mathcal{X}) \triangleq \log \mathbb{E}_{(x,y) \sim \mathcal{X}^2} \left[ e^{-t\|f(x) - f(y)\|_2^2} \right], \tag{2}$$

where $t > 0$ is a fixed parameter and is empirically set to $t = 2$. To note here, the vectors in the representation space are automatically $l_2$-normalized, i.e. $f(x) \triangleq f(x)/\|f(x)\|_2$, as we limit the representation space to a hypersphere following Wang & Isola (2020) and Grill et al. (2020) and the representation vectors in the following context are also automatically $l_2$-normalized, unless specified otherwise. Wang & Isola (2020) has empirically demonstrated that the balance of the alignment loss

and the uniformity loss is necessary when learning representations through contrastive method. The rationale behind it is straightforward: $\mathcal{L}_{\text{align}}$ provides the motive power that concentrates the similar data, and $\mathcal{L}_{\text{uniform}}$ prevents it from mapping all the data to the same meaningless point.

## 2.2 BYOL: BIZARRE ALTERNATIVE OF CONTRASTIVE

A recently proposed self-supervised representation learning algorithm BYOL hugely challenges the common understanding, that the alignment should be balanced by negative samples during the contrastive learning. It establishes two networks, online and target, approaching to each other during training. The online is trained to predict the target's representations and the target is the Exponential Moving Average (EMA) of the parameters of the online. The loss of BYOL at every iteration could be written as

$$\mathcal{L}_{\text{BYOL}} \triangleq \mathbb{E}_{(x,t_1,t_2)\sim(\mathcal{X},\mathcal{T}_1,\mathcal{T}_2)} \left[ \left\| q_w(f_\theta(t_1(x))) - f_\xi(t_2(x)) \right\|_2^2 \right], \tag{3}$$

where two vectors in representation space are automatically $l_2$-normalized. $f_\theta$ is the online encoder network parameterized by $\theta$ and $q_w$ is the predictor network parameterized by $w$. $x \sim \mathcal{X}$ is the input sampled from the data distribution $\mathcal{X}$, and $t_1(x)$, $t_2(x)$ are two augmented views of $x$ where $t_1 \sim \mathcal{T}_1, t_2 \sim \mathcal{T}_2$ are two data augmentations. The target network $f_\xi$ is of the same architecture as $f_\theta$ and is updated by EMA with $\tau$ controlling to what degree the target network preserves its history

$$\xi \leftarrow \tau\xi + (1-\tau)\theta. \tag{4}$$

From the scheme of BYOL training, it seems like there is no constraint on the uniformity, and thus most frequently asked question about BYOL is how it prevents the representation collapse. Theoretically, we would expect that when the final convergence of the online and target is reached, $\mathcal{L}_{\text{BYOL}}$ degenerates to $\mathcal{L}_{\text{align}}$ and therefore causes representation collapse, while this speculation never happens in reality. Despite the perfect SOTA performance of BYOL, there is one inconsistency not to be neglected: it fails with representation collapse when the predictor is removed, which means $q_w(x) = x$ for any given $x$. This inconsistent behavior of BYOL weakens its reliability and further poses questions on future adaptation of the algorithm. The motivation of understanding and even solving this inconsistency is the start point of this paper.

## 3 ON-AND-OFF BYOL: FAILURE WITHOUT PREDICTOR

We start by presenting a dissatisfactory property of BYOL: its success heavily relies on the existence of the predictor $q_w$. The experimental setup of this paper is listed in Appendix C. The performance of BYOL original model, whose predictor $q_w$ is a two-layer MLP with batch normalization, evaluated on the linear evaluation protocol (Kolesnikov et al., 2019; Kornblith et al., 2019; Chen et al., 2020a; He et al., 2020; Grill et al., 2020) reaches $68.08 \pm 0.84\%$. When the predictor is removed, the performance degenerates to $20.92 \pm 1.29\%$, which is even lower than the random baseline's $42.74 \pm 0.41\%$. We examine the speculation that the performance drop is caused by the representation collapse both visually (refer to Appendix F.1) and numerically. Inspired by Wang & Isola (2020), we use $\mathcal{L}_{\text{uniform}}(f_\theta; \mathcal{X})$ to evaluate to what degree the representations are spread on the hypersphere and $\mathcal{L}_{\text{align}}(q_w \circ f_\theta)$ to evaluate how the similar samples are aligned in the representation space. The results in Table 1 show that with the predictor, BYOL optimizes the uniformity of the representation distribution. On the contrary, when taken away the predictor, the alignment of two augmented views is overly optimized and the uniformity of the representation deteriorates (Figure 4), therefore we conclude the predictor is essential to the collapse prevention in BYOL.

One reasonable follow-up explanation on the efficacy of the predictor may consider its specially designed architecture or some good properties brought by the weight initialization, which makes it hard to understand the mechanism behind it. Fortunately, after replacing the current predictor, two-layer MLP with batch normalization (Ioffe & Szegedy, 2015), with different network architectures and weight initializations, we find that there is no significant change either on linear evaluation protocol or on the model behavior during training (Table 1, for detailed training trajectory, refer to Figure 4). We first replace the complex structure with linear mapping $q_w(\cdot) = W(\cdot)$. This replacement provides a naive solution to representation collapse: $W = I$, while it never converges to this apparent collapse. Surprisingly enough when we go harsher on this linear predictor by initializing

Table 1: Evaluation results of BYOL variants on CIFAR10 after 300 epochs of training, used as evidence supporting the proposal of RAFT. $(\alpha, \beta) = (1, 1)$ are set in BYOL$'$ and RAFT. $\mathcal{L}_{\text{align}}(q_w \circ f_\theta)$ and $\mathcal{L}_{\text{uniform}}(f_\theta)$ are evaluated by averaging the last 10 epochs of training. **We highlight the overly optimized $\mathcal{L}_{\text{align}}$ and failed $\mathcal{L}_{\text{uniform}}$ in this table.** For the accuracy on the linear evaluation protocol, we also **only highlight the ones that underperform the random baseline.**

| Model | $q_w$ | $\mathcal{L}_{\text{align}}(q_w \circ f_\theta)$ | $\mathcal{L}_{\text{uniform}}(f_\theta)$ | Linear Evaluation Protocol(%) |
|---|---|---|---|---|
| Rand-Baseline | $W$ | $78.09 \times 10^{-4}$ | $-0.51$ | $42.74 \pm 0.41$ |
| BYOL | MLP | $25.03 \times 10^{-4}$ | $-2.22$ | $64.32 \pm 0.89$ |
| BYOL$'$ | MLP | $22.09 \times 10^{-4}$ | $-2.07$ | $69.21 \pm 1.01$ |
| RAFT | MLP | $18.90 \times 10^{-4}$ | $-2.04$ | $71.31 \pm 0.75$ |
| BYOL-LP | $W$ | $7.71 \times 10^{-4}$ | $-2.16$ | $67.64 \pm 0.90$ |
| BYOL$'$-LP | $W$ | $7.32 \times 10^{-4}$ | $-2.19$ | $68.61 \pm 0.73$ |
| RAFT-LP | $W$ | $7.42 \times 10^{-4}$ | $-2.23$ | $67.55 \pm 0.55$ |
| BYOL-NP | $I$ | $\mathbf{1.94 \times 10^{-10}}$ | $\mathbf{-0.14}$ | $\mathbf{20.92 \pm 1.29}$ |
| BYOL$'$-NP | $I$ | $\mathbf{1.35 \times 10^{-10}}$ | $\mathbf{-0.10}$ | $\mathbf{16.92 \pm 1.05}$ |
| RAFT-NP | $I$ | $16.07 \times 10^{-4}$ | $\mathbf{-0.006}$ | $\mathbf{11.72 \pm 0.05}$ |
| TanBYOL-LP | $W$ | $7.83 \times 10^{-4}$ | $-1.92$ | $67.63 \pm 1.27$ |
| TanBYOL$'$-LP | $W$ | $7.52 \times 10^{-4}$ | $-2.19$ | $67.66 \pm 0.76$ |
| TanRAFT-LP | $W$ | $7.61 \times 10^{-4}$ | $-2.13$ | $69.31 \pm 0.87$ |

$W$ with the apparent collapse solution $I$, the model itself seems to have a self-recovering mechanism even though it starts off at a poor position: the loss quickly approaches to 0 and the uniformity deteriorates for 10-20 epochs and suddenly it deflects from the collapse and keeps on the right track. We conduct a theoretical proof that a randomly initialized linear predictor prevents the (more strict form of) representation collapse by creating infinite non-trivial solutions when the convergence is achieved (refer to Appendix I), while we fail to correlate the consistently optimized uniformity with the presence of the predictor, which indicates that a deeper rationale needs to be found.

## 4 RUN AWAY FROM YOUR TEACHER: MORE EFFECTIVE REGULARIZER

### 4.1 DISENTANGLE THE BYOL LOSS BY UPPER BOUNDING

Analyzing $\mathcal{L}_{\text{BYOL}}$ is hard, since it only has one single mean squared error term and there are many factors entangled within it, e.g., two augmented views of the same data, predictor, and the EMA updating rule. Inspired by the Bias-Variance decomposition on squared loss (Geman et al., 1992), we extract the alignment loss by subtracting and adding the same term $q_w(f_\theta(t_2(x)))$ and further yield the upper bound of $\mathcal{L}_{\text{BYOL}}$. For details, please refer to Appendix G.

**Definition 4.1 (Cross-model loss)** *The **cross-model loss** $\mathcal{L}_{\text{cross-model}}(f, g; \mathcal{X})$ of the function $f$ and $g$ over the data distribution $\mathcal{X}$ is defined as*

$$\mathcal{L}_{\text{cross-model}}(f, g; \mathcal{X}) \triangleq \mathbb{E}_{x \sim \mathcal{X}} \left[ \left\| f(x) - g(x) \right\|_2^2 \right]. \tag{5}$$

**Definition 4.2 (BYOL$'$ loss)** *The **BYOL$'$ loss** $\mathcal{L}_{\text{BYOL}'}$ is defined as*

$$\mathcal{L}_{\text{BYOL}'} \triangleq \alpha \mathcal{L}_{\text{align}}(q_w \circ f_\theta; \mathcal{P}_{\text{pos}}) + \beta \mathcal{L}_{\text{cross-model}}(q_w \circ f_\theta, f_\xi; \mathcal{X}_2) \tag{6}$$

where $\alpha, \beta > 0$ are constants, $\mathcal{P}_{\text{pos}}$ is defined in Eq. 1 and $\mathcal{X}_2 = \mathcal{T}_2(\mathcal{X})$ is the distribution of the augmented data. For the sake of simplicity, we use $\mathcal{L}_{\text{align}}(q_w \circ f_\theta)$ to denote $\mathcal{L}_{\text{align}}(q_w \circ f_\theta; \mathcal{P}_{\text{pos}})$ in the following content. For the sake of symmetry, we use $\mathcal{L}_{\text{cross-model}}(q_w \circ f_\theta)$ to denote $(1/2)[\mathcal{L}_{\text{cross-model}}(q_w \circ f_\theta, f_\xi, \mathcal{X}_1) + \mathcal{L}_{\text{cross-model}}(q_w \circ f_\theta, f_\xi, \mathcal{X}_2)]$ to compute the cross-model loss.

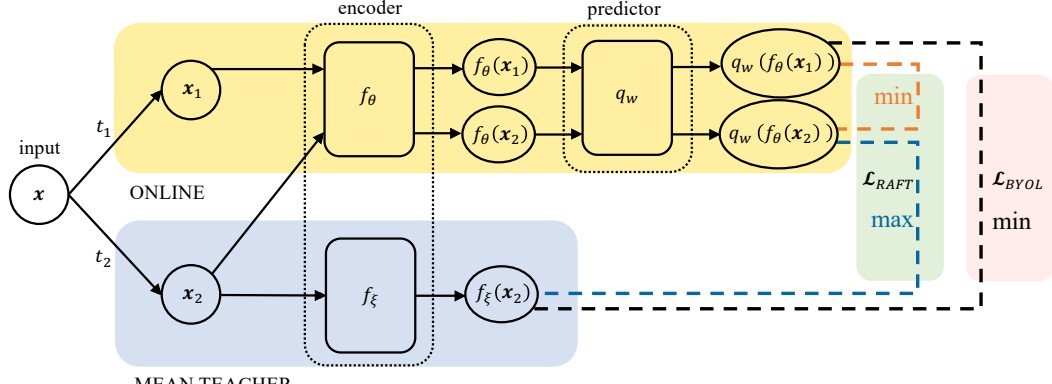

Figure 1: Framework diagram of RAFT and BYOL. The online network is composed of an encoder $f_\theta$ and an extra predictor $q_w$. The Mean Teacher $f_\xi$ is the EMA of the encoder $f_\theta$. In BYOL, the loss is computed by minimizing the distance between the prediction of one view $x_1$ and another view $x_2$'s representation generated by the MT. In RAFT, we optimize two objectives together: (i) minimize the representation distance between two samples from a positive pair and (ii) maximize the representation distance between the online network and its MT.

**Theorem 4.1 ($\mathcal{L}_{\mathrm{BYOL'}}$ is an upper bound of $\mathcal{L}_{\mathrm{BYOL}}$)** *$\mathcal{L}_{\mathrm{BYOL'}}$ is an upper bound of $\mathcal{L}_{\mathrm{BYOL}}$ if we ignore the scalar multiplication. Concretely speaking, for any given constants $\alpha, \beta > 0$, we have*

$$\mathcal{L}_{\mathrm{BYOL}} \leq (\frac{1}{\alpha} + \frac{1}{\beta})\mathcal{L}_{\mathrm{BYOL'}}. \tag{7}$$

**Proof** *Please refer to Appendix G.*

Ideally, minimizing $\mathcal{L}_{\mathrm{BYOL'}}$ would yield similar performance as minimizing $\mathcal{L}_{\mathrm{BYOL}}$. We exemplify the legitimacy of $\mathcal{L}_{\mathrm{BYOL'}}$ by setting $(\alpha, \beta) = (1, 1)$. In Table 1, the performance of BYOL and BYOL′ are close to each other with respect to three metrics: alignment, uniformity, and downstream linear evaluation protocol, regardless of the form of predictors. When the predictor is linear mapping, the performance differences between them are subtle. Besides, when the predictor is removed, the representation collapse also happens to BYOL′. So we conclude that optimizing $\mathcal{L}_{\mathrm{BYOL'}}$ is almost equivalent to $\mathcal{L}_{\mathrm{BYOL}}$. In spite of the performance similarity, $\mathcal{L}_{\mathrm{BYOL'}}$ is of a more disentangled form than $\mathcal{L}_{\mathrm{BYOL}}$ and therefore we focus on studying the former instead of the latter.

The new objective consists of two terms: the first term $\mathcal{L}_{\mathrm{align}}$ minimizes the representation distance between samples from a positive pair and has already been shown crucial to the successful contrastive methods (Wang & Isola, 2020). Intuitively, it provides the motive power to concentrate similar data in the representation space. Based on the form of BYOL′, we conclude that MT is used to regularize the alignment loss. This perspective of two terms regularizing each other is crucial to our analysis and improvement of the original BYOL framework. Understanding why BYOL works without collapse is approximately equivalent to understanding how minimizing $\mathcal{L}_{\mathrm{cross\text{-}model}}(q_w \circ f_\theta, f_\xi)$ effectively regularizes the alignment loss, or even actively optimizes the uniformity.

## 4.2 RAFT: RUN AWAY FROM YOUR TEACHER

The major difficulty of correlating $\mathcal{L}_{\mathrm{cross\text{-}model}}$ with $\mathcal{L}_{\mathrm{uniform}}$ is that their optimization intentions are not only irrelevant, but somewhat opposite. Minimizing the cross-model loss asks the network to produce close representations for certain inputs, while optimizing the uniformity loss requires it to produce varying representations. The disparity residing in the form pushes us to question the original motivation of BYOL: do we really want the online network to approach to the Mean Teacher? To test our suspicion, we minimize $[\mathcal{L}_{\mathrm{align}}(q_w \circ f_\theta) - \mathcal{L}_{\mathrm{cross\text{-}model}}(q_w \circ f_\theta, f_\xi)]$ instead of $[\mathcal{L}_{\mathrm{align}}(q_w \circ f_\theta) + \mathcal{L}_{\mathrm{cross\text{-}model}}(q_w \circ f_\theta, f_\xi)]$, and we find it works as well. This bizarre phenomenon will be explained in Section 5. Removing the predictor, we observe that although minimizing $[\mathcal{L}_{\mathrm{align}}(f_\theta) - \mathcal{L}_{\mathrm{cross\text{-}model}}(f_\theta, f_\xi)]$ fails to yield better representation than the random baseline,

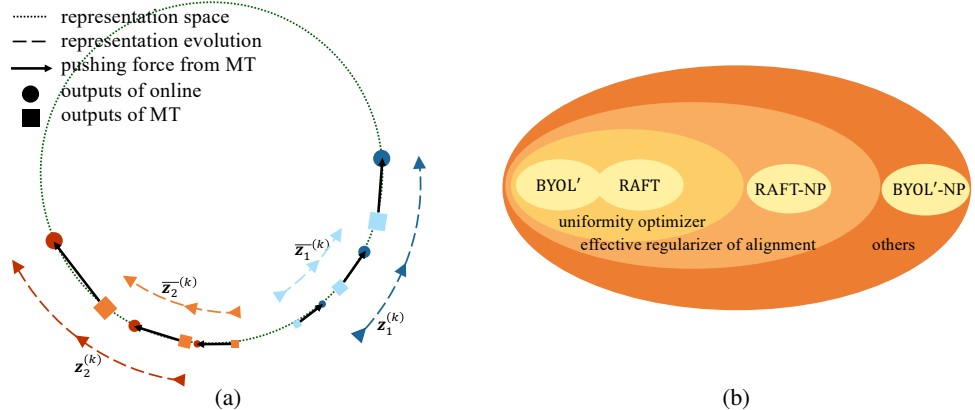

Figure 2: Analysis on the legitimacy of RAFT and why it's more favorable. **(a)** Diagram demonstrating how RAFT conceptually works: if two samples' updating directions are opposite, MT helps pushing them away at the next several iterations. Here $z_i = q_w\left(f_\theta(x_i)\right)$, $\overline{z_i} = f_\xi(x_i)$, $i = 1, 2$. **(b)** Objective categories diagram with respect to the effect constraining the alignment loss. In contrastive methods, the most favorable objective actively optimizes uniformity, including both BYOL' and RAFT when there exists predictor. The secondly favorable objective is the effective regularizer of alignment. RAFT remains to restrain alignment loss without predictor, while BYOL fails to do so, which implies that RAFT is a more unified objective.

it prevents the overly-optimized alignment loss, i.e. it works as an effective regularizer for the alignment loss, while minimizing $\mathcal{L}_{\text{cross-model}}(f_\theta, f_\xi)$ does not.

Based on the conclusion above and law of Occam's Razor, we propose a new self-supervised learning framework, Run Away From your Teacher (RAFT), which optimizes two learning objectives simultaneously: (i) minimize the alignment loss of two samples from a positive pair and (ii) maximize the distance between the online network and its MT (refer to Figure 1 and Algorithm 1).

**Definition 4.3 (RAFT loss)** *The **RAFT loss** $\mathcal{L}_{\text{RAFT}}$ is defined as*

$$\mathcal{L}_{\text{RAFT}} \triangleq \alpha\mathcal{L}_{\text{align}}(q_w \circ f_\theta; \mathcal{P}_{\text{pos}}) - \beta\mathcal{L}_{\text{cross-model}}(q_w \circ f_\theta, f_\xi; \mathcal{X}_2), \tag{8}$$

where $\alpha, \beta > 0$ are constants and other components follows the Definition 4.2.

Compared to BYOL and BYOL', RAFT better conforms to our knowledge and is a conceptually non-collapsing algorithm. There has been a lot of work demonstrating that weight averaging is roughly equal to sample averaging (Tarvainen & Valpola, 2017), thus if two samples' representations are close to each other at the beginning and their initial updating directions are opposite, then RAFT consistently separates them in the representation space. All the forms of loss terms could be classified into three categories: uniformity optimizer, effective regularizer for alignment loss, and others (refer to Figure 2b). According to our experiments, when the predictor is removed, running away from MT remains an effective regularizer for the alignment loss while BYOL's running towards MT fails to do so, thus RAFT is of more unified and consistent form. In summary, our proposed learning framework RAFT is completely based on the intention of solving the inconsistency of the predictor in BYOL, and it's better than BYOL in threefold:

- Consistency. Compared to BYOL, our newly proposed method has an effective regularizer for the alignment loss regardless of the presence of predictor.

- Interpretability. Mean teacher uses the technique of weight averaging and thus could be considered as an approximate ensemble of the previous versions of the model. Running away from the mean teacher intuitively encourages the diversity of the representation, which is positively correlated to the uniformity.

- Disentanglement. The learning objective is decoupled into aligning two augmented views and running away from the mean teacher, and hence could be independently studied.

We will discuss the relationship between RAFT and BYOL$'$ in the next section, and we find BYOL$'$ is a special form of RAFT under certain conditions, which makes the performance of BYOL$'$ a guarantee of the effectiveness of RAFT. We provide benchmarks of alignment, uniformity, and downstream linear evaluation performance on CIFAR10 (Table 3). We discover that balancing the alignment loss and the cross-model loss is not an easy job with the predictor taken away. The imbalance between the alignment loss and the cross-model loss would lead to representation collapse or over-regularized alignment where every data is randomly projected. One interesting research direction is to study the efficacy of the predictor. The reason why it helps the two terms to achieve an equilibrium is left to be answered.

## 5 UNDERSTANDING BYOL VIA RAFT

In Section 4.1 we derive an upper bound $\mathcal{L}_{\text{BYOL}'}$ of $\mathcal{L}_{\text{BYOL}}$ and explicitly extract two terms $\mathcal{L}_{\text{align}}$ and $\mathcal{L}_{\text{cross-model}}$. In BYOL$'$, two terms are simultaneously minimized, while in RAFT, we minimize $\mathcal{L}_{\text{align}}$ but maximize $\mathcal{L}_{\text{cross-model}}$ instead. To clearly distinguish the difference between the two objectives, we rewrite them as following:

$$\mathcal{L}_{\text{BYOL}'} = \alpha \mathcal{L}_{\text{align}}(q_w \circ f_\theta) + \beta \mathcal{L}_{\text{cross-model}}(q_w \circ f_\theta, f_\xi), \tag{9}$$

$$\mathcal{L}_{\text{RAFT}} = \alpha \mathcal{L}_{\text{align}}(q_w \circ f_\theta) - \beta \mathcal{L}_{\text{cross-model}}(q_w \circ f_\theta, f_\xi), \tag{10}$$

where $\alpha, \beta > 0$ are constants.

In form, $\mathcal{L}_{\text{RAFT}}$ and $\mathcal{L}_{\text{BYOL}'}$ seem to evolve in opposite optimizing direction on the second term, but the empirical study has shown that both of them work. How can two opposite optimization goals produce similar effect? Since RAFT is a conceptually working method, we analyze the mechanism of BYOL$'$ by establishing the equivalence between the parameters of BYOL$'$ and RAFT under mild conditions.

**Theorem 5.1 (One-to-one correspondence between BYOL$'$ and RAFT)** *There is a one-to-one correspondence between parameter trajectories of BYOL$'$ and RAFT when the following three conditions hold:*

  *i.   the representation space is a hypersphere;*

  *ii.  the predictor is a linear transformation, i.e. $q_w(\cdot) = W(\cdot)$;*

  *iii. only the tangential component of the gradient on the hypersphere is preserved.*

**Proof** *We prove the theorem by construction. For the detail, please refer to Appendix H.*

**Remark** The third condition conforms to the property of the hypersphere representation space and is easy to achieve. One can preserve only the tangential gradient by slightly modifying the loss. For example, suppose the representation of the MT is $\bar{z}$ and the representation of the input is $z$ which are both normalized, the cross-model loss $\left\| \bar{z} - z \right\|_2^2$ can be revised as $\left\| \bar{z} - \lambda z \right\|_2^2 / \lambda$, where $\lambda = \text{sg}(\langle z, \bar{z} \rangle)$ stands for stopping gradient of the inner product $\langle z, \bar{z} \rangle$. Our experiments in Table 1 demonstrates that the condition of the tangential component of the gradient doesn't turn any of the algorithms including BYOL, BYOL$'$ and RAFT into a collapsed one.

In Theorem 1, we show that optimizing $\mathcal{L}_{\text{BYOL}'}$ with initial parameters $(\theta^{(0)}, W^{(0)})$ is equivalent to optimizing $\mathcal{L}_{\text{RAFT}}$ with initial parameters $(\theta^{(0)}, -W^{(0)})$ when the aforementioned three conditions are satisfied. This equivalence demonstrates that the final encoder network $f_\theta$ and $f_{\theta'}$ equal to each other. Therefore we conclude that, as representation learning framework, BYOL$'$ is equivalent to our newly proposed RAFT.

From a geometric point of view, the optimization process is the data points moving in the representation space under the guidance of the training loss. The loss function measures the potential energy of the parameters, and the gradient with regard to the data points is the motive force. If the representation space is a hypersphere as in BYOL, then the tangential force, i.e. the tangential component of the gradient, is the only key to scattering or concentrating the data points in the representation space. By the central symmetry of the hypersphere, clockwise and counterclockwise moving directions are equivalent to some extent, for example, pushing a point by $\pi/2$ and pulling it by $\pi/2$ on the 2-dimensional sphere causes the same effect.

The equivalence between BYOL$'$ and RAFT offers us a direct way to understand some strange phenomena we observe which are also reported in the original BYOL paper. Firstly, the non-collapse of BYOL is explained, since the RAFT is an intuitively and practically working algorithm. The equivalence of BYOL$'$ and RAFT when predictor is linear helps us understand why BYOL is an effective self-supervised learning algorithm. It also explains our initial question why BYOL fails to avoid representation collapse without the predictor: removing the predictor means fixing $W = I$, which breaks the RAFT's designing principle of running away from the MT. Secondly, though the BYOL's optimization procedure is of the form that two models approaching to each other, there has been no report of convergence in the original paper. The established equivalence perfectly explains it. RAFT incorporates the MT in an extremely dynamic way since it continuously varies from the history models, thus there would be no convergence of the data points. So does the parameters.

## 6 CONCLUSION AND FUTURE WORK

In this paper, we address the problem of why the newly proposed self-supervised learning framework Bootstrap Your Own Latent (BYOL) works without negative samples. By decomposing, upper bounding and approximating the original loss of BYOL, we establish another interpretable self-supervised learning method, Run Away From your Teacher. We show that RAFT contains an explicit term that prevents the representation collapse and we also empirically validate the effectiveness of RAFT. By constructing a one-to-one correspondence from RAFT to BYOL$'$ (variant of BYOL), we successfully explain the mechanism behind BYOL that makes it work and therefore implies the huge potential of our proposed RAFT. Based on the observation and the conclusion, here we have several suggestions for future work:

**Theoretical guarantees of RAFT.**  Though we have intuitively explained why running away from the MT is an effective regularizer, we don't provide theoretical guarantees why optimizing RAFT would be favorable with respect to the representation learning. In future, one can try to relate RAFT to the theory of Mutual Information (MI) maximization (Belghazi et al., 2018; Hjelm et al., 2018; Tschannen et al., 2019), as the training objective of contrastive learning InfoNCE has been proven to be a lower bound of the MI (Poole et al., 2019). One detail should be noticed when attempting to correlate RAFT with MI maximization. Even though RAFT is an effective regularizer, it fails to yield good-quality representations when the predictor is removed, thus any theoretical proof on the effectiveness of RAFT should well explain the mechanism behind this extra predictor.

**On the efficacy of the predictor.**  It has become a popular and almost standardized method to add an extra MLP on top of the network in contrastive learning methods (Chen et al., 2020a;b; Grill et al., 2020), while most of the work adopts this method as a special trick without considering the effect this MLP brings to the algorithm. In this paper, however, we find that this extra MLP may bring some unexpected properties to the original training objective: although the representations are optimized by disparate motivations (in our paper, BYOL$'$ running towards MT and RAFT running away from MT), the encoder network is trained to be exactly the same. This observation indicates that the mechanism of the extra MLP to the network needs to be further studied.

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

## A  MAIN THREAD OF PAPER

The proposal of RAFT is based on a series of theoretical derivation and empirical approximation. Therefore the logic chain of our paper is fundamental to the legitimacy of our explanation on BYOL and the superiority of our newly proposed RAFT. Here we organize our main thread in the same order as the sections, to provide a clear view with readers.

In Section 3,

- As a learning framework, BYOL does not consistently work. It heavily relies on the existence of the predictor. We want to understand why this inconsistency exists.

- The architecture of the predictor doesn't affect the collapse of BYOL, the fact that the linear predictor $q_w(\cdot) = W(\cdot)$ prevents collapse will be used as a crucial condition in Section 5.

In Section 4.1,

- A new disentangled objective $\mathcal{L}_{\mathrm{BYOL'}} = \alpha \mathcal{L}_{\mathrm{align}}(q_w \circ f_\theta) + \beta \mathcal{L}_{\mathrm{cross\text{-}model}}(q_w \circ f_\theta, f_\xi)$ is established by upper bounding.

- We showcase that minimizing $\mathcal{L}_{\mathrm{BYOL}}{'}$ is close to minimizing $\mathcal{L}_{\mathrm{BYOL}}$ in terms of alignment, uniformity, and linear evaluation protocol, which indicates that understanding the behavior of optimizing BYOL's upper bound is approximately equivalent to understanding BYOL.

In Section 4.2,

- We find minimizing $[\mathcal{L}_{\mathrm{align}}(q_w \circ f_\theta) - \mathcal{L}_{\mathrm{cross\text{-}model}}(q_w \circ f_\theta, f_\xi)]$ works as well, which has the exact opposite way of incorporating the cross-model loss to BYOL$'$.

- Based on the observation above, we propose a new self-supervised learning approach Run Away From your Teacher, which regularizes $\mathcal{L}_{\mathrm{align}}(q_w \circ f_\theta)$ by maximizing $\mathcal{L}_{\mathrm{cross\text{-}model}}(q_w \circ f_\theta, f_\xi)$. Compared with BYOL, RAFT accords more with our common understanding.

- Additional experiments show that without predictor, BYOL$'$ fails to regularize $\mathcal{L}_{\mathrm{align}}(f_\theta)$, let alone optimizing uniformity. On the contrary, although not able to actively optimize uniformity either, RAFT's maximizing $\mathcal{L}_{\mathrm{cross\text{-}model}}(f_\theta, f_\xi)$ continues to be an effective regularizer for $\mathcal{L}_{\mathrm{align}}(f_\theta)$, which makes it more favorable (Figure 2b).

In Section 5,

- We prove that when the predictor is linear ($q_w = W$) and the representation space is a hypersphere where only the tangential component of gradient is preserved during training, minimizing $\mathcal{L}_{\mathrm{cross\text{-}model}}(W \circ f_\theta, f_\xi)$ and maximizing it obtain the same encoder $f_\theta$.

- Based on the equivalence above, we conclude that BYOL$'$ is a special case of RAFT under conditions above. The equivalence established helps understanding several counter-intuitive behaviors of BYOL.

# B    BACKGROUND AND RELATED WORK

## B.1    CONTRASTIVE LEARNING

Contrastive methods relies on the assumption that two views of the same data point share the information, and thus creates a positive pair. By separating the positives and the negatives, the neural network trained by the algorithm learns to extract the most useful information from the data and performs better on the downstream tasks. Typically, the algorithm uses the InfoNCE objective:

$$\mathcal{L}_{\text{contrast}}(h, K) = \mathop{\mathbb{E}}_{\substack{(x,x^+)\sim\mathcal{P}_{\text{pos}} \\ \{x_i^-\}_{i=1}^K\sim\mathcal{X}^K}} \left[ -\log \frac{e^{h(x,x^+)}}{e^{h(x,x^+)} + \sum_{i=1}^K e^{h(x,x_i^-)}} \right], \tag{11}$$

where $(x, x^+)$ are sampled from the positive pair distribution $\mathcal{P}_{\text{pos}}$, which is built by a series of data augmentation functions [ref]. The negative samples $\{x_i^-\}^K$ are i.i.d sampled for $K$ times from the data distribution $\mathcal{X}$; function $h(x, y)$ measures the similarity between two input data $(x, y)$. Empirically for the sake of symmetry, the measurement function $h(x, y) = d(f(x), f(y))$ has an encoder $f(\cdot)$ and a similarity metric $d(\cdot, \cdot)$ evaluating how close the two representations are.

## B.2    MEAN TEACHER

There is one type of semi-supervised learning method that BYOL constantly reminds people of, Mean Teacher (MT) (Tarvainen & Valpola, 2017; Laine & Aila, 2016). Like BYOL, MT is also of Teacher-Student (T-S) framework, where the teacher network is also the EMA of the student network. The additional consistency loss between the teacher and student is applied to the supervised signals. There has been a lot of work demonstrating the efficacy of MT(Athiwaratkun et al., 2019; Novak et al., 2018; Chaudhari et al., 2019), among which the major conclusion states that the consistency loss between the student and its MT acts as a regularizer for better generalization. The proven properties of MT might lead us to focus on how the online network's learning from MT effectively regularizes $\mathcal{L}_{\text{align}}$ in BYOL. In this paper, however, we propose the opposite way of leveraging MT in contrastive methods.

# C    EXPERIMENTAL SETUP

**Dataset**    Our main goal is to unravel the mystery why BYOL doesn't collapse during training and to solve the predictor-inconsistency. The most important metric is whether the algorithm collapses or not, and we don't target on developing a more powerful self-supervised learning algorithm that surpasses SOTA on large dataset. In this repsect, we limit our experiments to the scope of the CIFAR10 dataset. Each image is resized from $32 \times 32$ to $96 \times 96$. This change is the consequence of the tradeoff between the effect of the data augmentation and batch size: larger size of the image would allow more subtle and informative data augmentation scheme while it will reduce the training batch size, which has already been empirically shown is harmful to the model performance.

**Model architecture**    In our experiments, the model is composed of three stages: an encoder $f_\theta$ that adopts the ResNet18 architecture (without the classifier on top); a projector $g_\theta$ that is comprised of a linear layer with output size 512, batch normalization, rectified linear units (ReLU), and a final linear layer with output size 128; a predictor $q_w$ that is comprised of the same architecture as the projector but without the batch normalization.

**Training**    We adopt the same data augmentation scheme that is used in Chen et al. (2020a) and Grill et al. (2020) and train the BYOL on the training set for 300 epochs with batch size 128 on 3 random seeds. The objective of training is specified accordingly and the model is trained on the Adam optimizer with learning rate $3 \times 10^{-4}$ (Kingma & Ba, 2014). Unless stated otherwise, we update the target network with the EMA rate $4 \times 10^{-3}$ without the cosine smoothing trick.

**Evaluation**    After training, we evaluate the encoder's performance on the widely adopted linear evaluation protocol: we fix the parameter of the encoder and we train another linear classifier on top of it using all the training labels for 100 epochs with learning rate $5 \times 10^{-4}$. The final classification accuracy indicates to what degree the representations of the same class concentrate and the representations of the different class separate, and thus tells the quality of the representation.

# D   TABLES

Table 2: Look-up table for the models that appear in the paper.

| Model Name | Description |
|---|---|
| BYOL-MLPP (BYOL) | **BYOL** with **MLP-P**redictor |
| BYOL-NP | **BYOL** with **N**o **P**redictor |
| BYOL-LP | **BYOL** with **L**inear **P**redictor |
| BYOL-LPI | **BYOL** with **L**inear **P**redictor initialized with $I$ |
| TanBYOL-LP | **BYOL** with **L**inear **P**redictor, preserving only the **Tan**gential gradient |
| | |
| BYOL′-MLPP (BYOL′) | trained with $\mathcal{L}_{\text{BYOL}'} = \mathcal{L}_{\text{align}} + \mathcal{L}_{\text{cross-model}}$, **MLP P**redictor |
| BYOL′-LP | trained with $\mathcal{L}_{\text{BYOL}'} = \mathcal{L}_{\text{align}} + \mathcal{L}_{\text{cross-model}}$, linear predictor |
| BYOL′-NP | trained with $\mathcal{L}_{\text{BYOL}'} = \mathcal{L}_{\text{align}} + \mathcal{L}_{\text{cross-model}}$, **N**o **P**redictor |
| TanBYOL′-LP | **BYOL**′ with **L**inear **P**redictor, preserving only the **Tan**gential gradient |
| | |
| RAFT-MLPP (RAFT) | **RAFT** with **MLP-P**redictor |
| RAFT-NP | **RAFT** with **N**o **P**redictor |
| RAFT-LP | **RAFT** with **L**inear **P**redictor |
| TanRAFT-LP | **RAFT** with **L**inear **P**redictor, preserving only the **Tan**gential gradient |

Table 3: Evaluation results of RAFT on CIFAR10. $\alpha$ and $\beta$ represents the weight of $\mathcal{L}_{\text{align}}(q_w \circ f_\theta)$, and $\mathcal{L}_{\text{cross-model}}(q_w \circ f_\theta, f_\xi)$, i.e. $\mathcal{L} = \alpha \mathcal{L}_{\text{align}}(q_w \circ f_\theta) + \beta \mathcal{L}_{\text{cross-model}}(q_w \circ f_\theta, f_\xi)$. All the quantifiable metrics are evaluated after 300 epochs of training of the training set of CIFAR10. Compared to BYOL′, our proposed RAFT is better in terms of the effectiveness of regularizing the $\mathcal{L}_{\text{align}}$.

| **Model** | $q_w$ | $\alpha$ | $\beta$ | $\mathcal{L}_{\text{align}}(q_w \circ f_\theta)$ | $\mathcal{L}_{\text{uniform}}(f_\theta)$ | Linear Evaluation Protocol(%) |
|---|---|---|---|---|---|---|
| Rand-Baseline | $W$ | - | - | $7.81 \times 10^{-3}$ | $-0.51$ | $42.74 \pm 0.41$ |
| RAFT | MLP | 1 | -0.1 | $8.37 \times 10^{-5}$ | $-2.00$ | $65.53 \pm 0.99$ |
| | MLP | 1 | -1 | $1.89 \times 10^{-3}$ | $-2.04$ | $71.31 \pm 0.75$ |
| | MLP | 1 | -10 | $1.00 \times 10^{-2}$ | $-0.29$ | $25.88 \pm 0.42$ |
| RAFT-LP | $W$ | 1 | -0.1 | $8.22 \times 10^{-6}$ | $-1.61$ | $52.57 \pm 2.72$ |
| | $W$ | 1 | -1 | $7.42 \times 10^{-4}$ | $-2.25$ | $67.55 \pm 0.55$ |
| | $W$ | 1 | -10 | $3.70 \times 10^{-4}$ | $-2.15$ | $66.10 \pm 0.82$ |
| RAFT-NP | $I$ | 1 | -1 | $1.61 \times 10^{-3}$ | $-0.01$ | $11.72 \pm 0.05$ |
| | $I$ | 1 | -10 | $1.54 \times 10^{-2}$ | $-0.99$ | $32.13 \pm 0.52$ |
| | $I$ | 1 | -100 | $1.56 \times 10^{-2}$ | $-1.29$ | $29.36 \pm 0.53$ |
| BYOL′-NP | $I$ | 1 | 1 | $1.35 \times 10^{-10}$ | $-0.12$ | $16.92 \pm 1.05$ |
| | $I$ | 1 | 10 | $2.38 \times 10^{-10}$ | $-0.88$ | $24.42 \pm 1.14$ |
| | $I$ | 1 | 100 | $4.55 \times 10^{-8}$ | $-1.14$ | $37.48 \pm 2.06$ |

# E    ALGORITHMS

---

**Algorithm 1:** RAFT: Run Away From your Teacher

**Inputs :**
    $\mathcal{X}, \mathcal{T}_1$, and $\mathcal{T}_2$                  set of images and distributions of transformations
    $\theta$ and $f_\theta$                        model parameters and encoder
    $w$ and $q_w$                        predictor parameters and predictor
    $\xi$ and $f_\xi$                         MT parameters and MT
    optimizer                      optimizer, updates online parameters using the loss gradient
    $K$ and $N$                       total number of optimization steps and batch size
    $\{\tau_k\}_{k=1}^K$ and $\{\eta_k\}_{k=1}^K$       target network update schedule and learning rate schedule

1   **for** $k = 1$ **to** $K$ **do**
2     $\mathcal{B} \leftarrow \{x_i\}_{i=1}^N \sim \mathcal{X}^N$            `// sample a batch of N images`
3     **for** $x_i \in \mathcal{B}$ **do**
4       $t_1 \sim \mathcal{T}_1$ and $t_2 \sim \mathcal{T}_2$        `// sample image transformations`
5       $z_1 \leftarrow q_w(f_\theta(t_1(x_i)))$ and $z_2 \leftarrow q_w(f_\theta(t_2(x_i)))$    `// reps for model`
6       $z_1' \leftarrow f_\xi(t_1(x_i))$ and $z_2' \leftarrow f_\xi(t_2(x_i))$      `// reps for MT`
7       $l_i = \left\| \frac{z_1}{\|z_1\|_2} - \frac{z_2}{\|z_2\|_2} \right\|_2^2$        `// loss for alignment`
8       $l_i' = -\frac{1}{2}\left( \left\| \frac{z_1}{\|z_1\|_2} - \frac{z_1'}{\|z_1'\|_2} \right\|_2^2 + \left\| \frac{z_2}{\|z_2\|_2} - \frac{z_2'}{\|z_2'\|_2} \right\|_2^2 \right)$   `// loss for cross-model`
9     **end**
10     $\delta\theta \leftarrow \frac{1}{N}\left( \sum_{i=1}^N \partial_\theta l_i + \partial_\theta l_i' \right)$      `// compute the loss gradient w.r.t.` $\theta$
11     $\theta \leftarrow \text{optimizer}(\theta, \delta\theta, \eta_k)$         `// update trainable parameters`
12     $\xi \leftarrow \tau_k\xi + (1 - \tau_k)\theta$          `// update target parameters`
13 **end**
    **Output:** encoder $f_\theta$

---

# F  VISUALIZATION OF TRAINING EVOLUTION

## F.1  REPRESENTATION DISTRIBUTION EVOLUTION OF DIFFERENT LEARNING ALGORITHMS

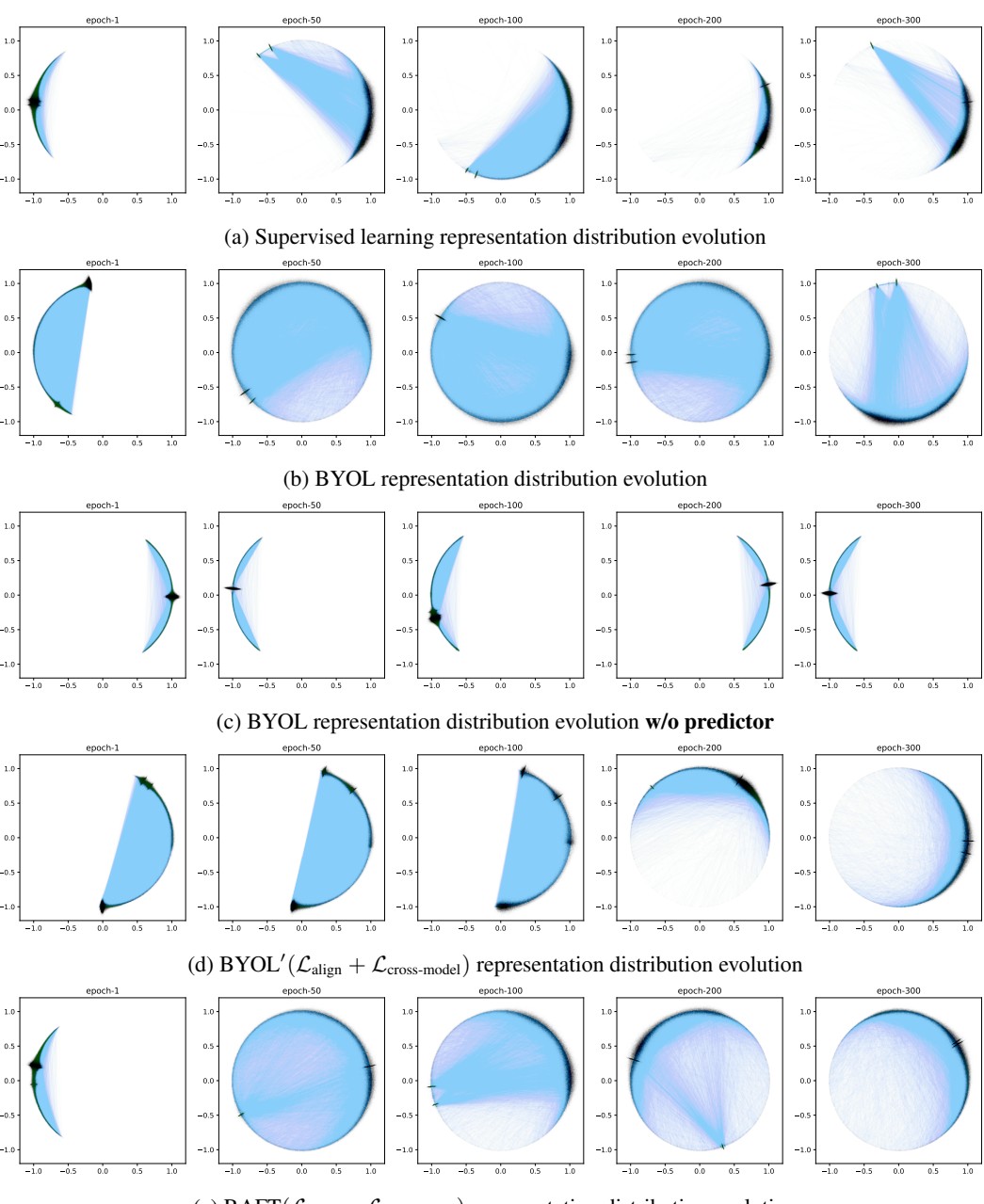

Figure 3: Visualization of the representation distribution evolution on CIFAR10 training set. We project the representation $f_\theta(x)$ to 2-D dimension using PCA (Wold et al., 1987) and then normalize to a unit sphere. The width of the circle shows the density of the data points that are projected to that particular position. Two dots residing on each side of the blue line across the circle represents two augmented views of the same data. **(a)** Supervised learning has no restriction on the uniformity of representation space. **(b)** BYOL with predictor evenly projects the data to different positions. **(c)** BYOL **w/o predictor** tends to project the huge portion of the data to the same position. **(d)** BYOL's upper bound BYOL′ also effectively disperses representations on the sphere. **(e)** Our RAFT shows that minimizing/maximizing $\mathcal{L}_{\text{cross-model}}$ has similar effect on the final representation distribution.

### F.2 EVOLUTION OF NUMERICAL METRIC IN DIFFERENT LEARNING ALGORITHMS

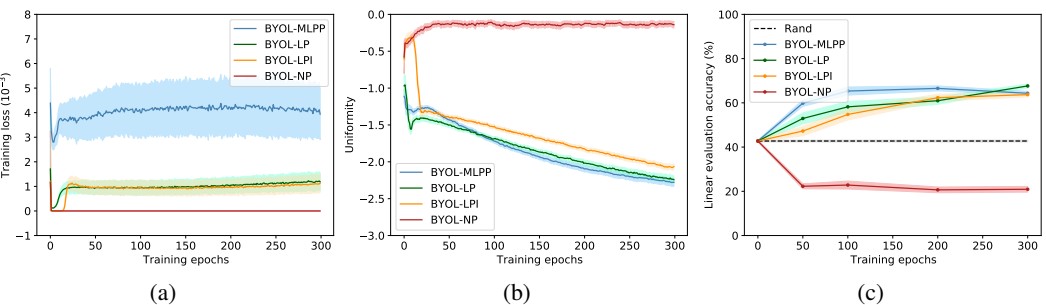

Figure 4: Training behaviors of BYOL with varying structures of the predictor. For detailed explanation on the model architecture, refer to Appendix C. **(a)** Evolution of training loss $\mathcal{L}_{\text{BYOL}}$. When taken out the predictor, the training loss quickly converges to 0 (red curve, BYOL-NP). Replacing the MLP-predictor (blue curve, BYOL-MLPP) with the linear predictor (green curve, BYOL-LP) will not cause the collapse even though $I$ is an apparent solution for collapse. Furthermore, initializing the linear predictor with $I$ forces the loss quickly approaching to 0 at beginning, while it recovers from the seemingly collapse after 10-20 epochs of training (orange curve, BYOL-LPI). **(b)** Evolution of the representation uniformity. BYOL with predictor consistently optimizes the uniformity of the representation distribution even though the uniformity is not explicitly included in the loss term. One interesting fact to note here is that the uniformity loss is optimized with a constant rate with linear predictor (green curve, BYOL-LP; orange curve, BYOL-LPI) after certain phase of training. **(c)** linear evaluation protocol on CIFAR10. Different structures of the predictor provide close performance on the downstream classification task.

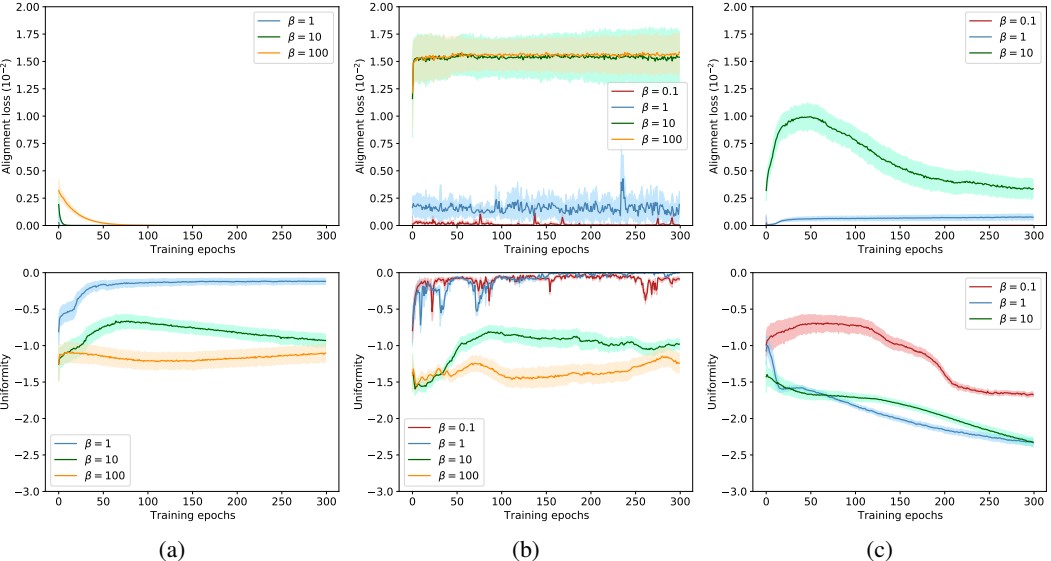

Figure 5: The evolution traces of $\mathcal{L}_{\text{align}}(q_w \circ f_\theta, f_\xi)$ and $\mathcal{L}_{\text{uniform}}(f_\theta)$ in BYOL' and our proposed RAFT. **(a)** Evolution trace of BYOL'-NP. Increasing $\beta$ (weight of $\mathcal{L}_{\text{cross-model}}$, regularizer for $\mathcal{L}_{\text{align}}$) does not prevent the failed regularization: $\mathcal{L}_{\text{align}}$ converges to 0 quickly. **(b)** Evolution trace of RAFT-NP. Small value of $\beta$ doesn't effectively regularize $\mathcal{L}_{\text{align}}$, but increasing the weight helps. In this respect, RAFT is a more effective regularizer, while the uniformity optimization holds no huge difference from BYOL'-NP. **(c)** Evolution trace of RAFT-LP. With the linear predictor, on the contrary to RAFT-NP, the uniformity is optimized consistently during training, which implies deeper rationale of the existence of predictor.

# G  PROOF OF BYOL UPPER BOUNDING

In this section, we provide how we derive the upper bound of $\mathcal{L}_{\text{BYOL}}$. For the sake of simplicity, without loss of rigor, we use $\boldsymbol{t}_1 = t_1(x)$ to represent the transformed input $x$.

$$\mathcal{L}_{\text{BYOL}} = \mathbb{E}_{x \sim \mathcal{X}, t_1 \sim \mathcal{T}_1, t_2 \sim \mathcal{T}_2} \left[ \left\| q_w(f_\theta(t_1(x))) - f_\xi(t_2(x)) \right\|_2^2 \right]$$

$$= \mathbb{E} \left[ \left\| q_w(f_\theta(x_1)) - q_w(f_\theta(x_2)) + q_w(f_\theta(x_2)) - f_\xi(x_2) \right\|_2^2 \right]. \tag{12}$$

By applying the Cauchy-Schwarz's inequality to Eq. 12, we yield:

$$\mathcal{L}_{\text{BYOL}} \le (1 + \frac{1}{\lambda}) \left( \mathbb{E} \left[ \left\| q_w(f_\theta(x_1)) - q_w(f_\theta(x_2)) \right\|_2^2 \right] + \lambda \mathbb{E} \left[ \left\| q_w(f_\theta(x_2)) - f_\xi(x_2) \right\|_2^2 \right] \right)$$

$$= (1 + \frac{1}{\lambda}) \left( \mathcal{L}_{\text{align}}(q_w \circ f_\theta; \mathcal{P}_{\text{pos}}) + \lambda \mathcal{L}_{\text{cross-model}}(q_w \circ f_\theta, f_\xi, \mathcal{X}) \right) \tag{13}$$

which stands for any $\lambda > 0$, where the positive-pair distribution $\mathcal{P}_{\text{pos}}$ is modeled by the chain rule of the conditional probability:

$$\mathcal{P}_{\text{pos}}(x_1, x_2) = \mathcal{X}(x) \cdot \mathcal{T}_1(t_1|x) \cdot \mathcal{T}_2(t_2|x).$$

For any given pair $\alpha, \beta > 0$, we let $\lambda = \beta/\alpha$ and substitute it back to Eq. 13, yielding

$$\mathcal{L}_{\text{BYOL}} \le (1 + \frac{\alpha}{\beta}) \left[ \mathcal{L}_{\text{align}}(q_w \circ f_\theta; \mathcal{P}_{\text{pos}}) + \frac{\beta}{\alpha} \mathcal{L}_{\text{cross-model}}(q_w \circ f_\theta, f_\xi, \mathcal{X}) \right]$$

$$= (\frac{1}{\alpha} + \frac{1}{\beta}) \left[ \alpha \mathcal{L}_{\text{align}}(q_w \circ f_\theta; \mathcal{P}_{\text{pos}}) + \beta \mathcal{L}_{\text{cross-model}}(q_w \circ f_\theta, f_\xi, \mathcal{X}) \right]$$

$$= (\frac{1}{\alpha} + \frac{1}{\beta}) \mathcal{L}_{\text{BYOL}}', \tag{14}$$

and as an optimization objective, we have

$$\min(\frac{1}{\alpha} + \frac{1}{\beta}) \mathcal{L}_{\text{BYOL}}' \Leftrightarrow \min \mathcal{L}_{\text{BYOL}}' \tag{15}$$

Therefore we have proven that $\mathcal{L}_{\text{BYOL}}'$ as optimization objective is the upper bound of $\mathcal{L}_{\text{BYOL}}$.

To note here, one can subtract and add a different term $f_\xi(x_1)$ to form the alignment loss on the side of MT $f_\xi$,

$$\mathcal{L}_{\text{BYOL}} = \mathbb{E} \left[ \left\| q_w(f_\theta(x_1)) - f_\xi(x_1) + f_\xi(x_1) - f_\xi(x_2) \right\|_2^2 \right], \tag{16}$$

while it doesn't help to solve the problem since the alignment constraint on the side of MT doesn't generate gradients.

# H  Proof of one-to-one correspondence between BYOL$'$ and RAFT

**Theorem (One-to-one correspondence between BYOL$'$ and RAFT)** *There is a one-to-one correspondence between parameter trajectories of BYOL$'$ and RAFT when the following three conditions hold:*

    *i.   the representation space is a hypersphere;*

    *ii.  the predictor is a linear transformation, i.e. $q_w(\cdot) = W(\cdot)$;*

    *iii. only the tangential component of the gradient on the hypersphere is preserved.*

Without losing generality, suppose that $x_1 = t_1(x), x_2 = t_2(x)$ where $x$ is an arbitrary input and batch size is 1, and $(\alpha, \beta) = (1, 1)$. We set BYOL$'$ and RAFT with initial parameters $(\theta', W') = (\theta^{(0)}, W^{(0)})$ and $(\theta, W) = (\theta^{(0)}, -W^{(0)})$ respectively. For convenience, we assumes the dot product "$\cdot$" ignores the row layout or column layout in the chain rule of derivatives and we define the following symbols:

$$\overline{z_2} = f_\xi(x_2), \tag{17}$$
$$z_1' = W' f_{\theta'}(x_1), \tag{18}$$
$$z_2' = W' f_{\theta'}(x_2), \tag{19}$$
$$z_1 = W f_\theta(x_1), \tag{20}$$
$$z_2 = W f_\theta(x_2). \tag{21}$$

Based on the notations defined, we rewrite the loss terms of BYOL$'$ and RAFT as follows:

$$\mathcal{L}_{\text{align}}^{\text{BYOL}'} = \left\| z_1' - z_2' \right\|_2^2, \tag{22}$$
$$\mathcal{L}_{\text{cross-model}}^{\text{BYOL}'} = \left\| z_2' - \overline{z_2} \right\|_2^2, \tag{23}$$
$$\mathcal{L}_{\text{align}}^{\text{RAFT}} = \left\| z_1 - z_2 \right\|_2^2, \tag{24}$$
$$\mathcal{L}_{\text{cross-model}}^{\text{RAFT}} = -\left\| z_2 - \overline{z_2} \right\|_2^2. \tag{25}$$

The two objectives are following:

$$\mathcal{L}_{\text{BYOL}'} = \mathcal{L}_{\text{align}}^{\text{BYOL}'} + \mathcal{L}_{\text{cross-model}}^{\text{BYOL}'}, \tag{26}$$
$$\mathcal{L}_{\text{RAFT}} = \mathcal{L}_{\text{align}}^{\text{RAFT}} + \mathcal{L}_{\text{cross-model}}^{\text{RAFT}}. \tag{27}$$

We claim that under the third condition, the following equations hold:

$$\left[\frac{\partial \mathcal{L}_{\text{BYOL}'}}{\partial \theta'}\right]_\| = \left[\frac{\partial \mathcal{L}_{\text{RAFT}}}{\partial \theta}\right]_\|, \left[\frac{\partial \mathcal{L}_{\text{BYOL}'}}{\partial W'}\right]_\| = -\left[\frac{\partial \mathcal{L}_{\text{RAFT}}}{\partial W}\right]_\|, \tag{28}$$

where subscript $\|$ denotes the tangential component of the gradient.

Firstly we show the equivalence with respect to $\theta$. Differentiate $\mathcal{L}_{\text{align}}^{\text{BYOL}'}, \mathcal{L}_{\text{align}}^{\text{RAFT}}$ with respect to $\theta'_{ij}$, $\theta_{ij}$ respectively, we obtain

$$\frac{\partial \mathcal{L}_{\text{align}}^{\text{BYOL}'}}{\partial \theta'_{ij}} = 2\left[(z_1' - z_2')_\| + (z_1' - z_2')_\perp\right] \cdot \left(\frac{\partial z_1'}{\partial \theta'_{ij}} - \frac{\partial z_2'}{\partial \theta'_{ij}}\right), \tag{29}$$

$$\frac{\partial \mathcal{L}_{\text{align}}^{\text{RAFT}}}{\partial \theta_{ij}} = 2\left[(z_1 - z_2)_\| + (z_1 - z_2)_\perp\right] \cdot \left(\frac{\partial z_1}{\partial \theta_{ij}} - \frac{\partial z_2}{\partial \theta_{ij}}\right), \tag{30}$$

$$\left[\frac{\partial \mathcal{L}_{\text{align}}^{\text{BYOL}'}}{\partial \theta'_{ij}}\right]_\| = 2\,(z_1' - z_2')_\| \cdot \left(\frac{\partial z_1'}{\partial \theta'_{ij}} - \frac{\partial z_2'}{\partial \theta'_{ij}}\right), \tag{31}$$

$$\left[\frac{\partial \mathcal{L}_{\text{align}}^{\text{RAFT}}}{\partial \theta_{ij}}\right]_\| = 2\,(z_1 - z_2)_\| \cdot \left(\frac{\partial z_1}{\partial \theta_{ij}} - \frac{\partial z_2}{\partial \theta_{ij}}\right), \tag{32}$$

where $(z_1' - z_2')$, $(z_1 - z_2)$ are vectors at the points $z_2'$ and $z_2$ on the hypersphere and we decompose the vector into the tangential (denoted by $\parallel$) and normal component (denoted by $\perp$):

$$(z_1' - z_2') = \left[ (z_1' - z_2')_{\parallel} + (z_1' - z_2')_{\perp} \right], (z_1 - z_2) = \left[ (z_1 - z_2)_{\parallel} + (z_1 - z_2)_{\perp} \right], \tag{33}$$

Generally, suppose $z$ is a unit vector starting at the origin point, which is perpendicular to the unit hypersphere at the point $z$, for any vector $\boldsymbol{v}$ starting at the point $z$, we have

$$\boldsymbol{v}_{\perp} = \langle \boldsymbol{v}, z \rangle \cdot z, \quad \boldsymbol{v}_{\parallel} = \boldsymbol{v} - \boldsymbol{v}_{\perp} = \boldsymbol{v} - \langle \boldsymbol{v}, z \rangle \cdot z. \tag{34}$$

Then we can compute the tangential component of the gradient:

$$
\begin{aligned}
(z_1' - z_2')_{\parallel} &= (z_1' - z_2') - \langle z_1' - z_2', z_2' \rangle \cdot z_2' \\
&= z_1' - \langle z_2', z_1' \rangle \cdot z_2', \\
(z_1 - z_2)_{\parallel} &= (z_1 - z_2) - \langle z_1 - z_2, z_2 \rangle \cdot z_2 \\
&= z_1 - \langle z_2, z_1 \rangle \cdot z_2.
\end{aligned}
\tag{35}
$$

Because of the initialization, $z_1' = -z_1$, $z_2 = -z_2'$, therefore we have

$$(z_1' - z_2')_{\parallel} = -(z_1 - z_2)_{\parallel}, \tag{36}$$

$$\left( \frac{\partial z_1'}{\partial \theta_{ij}'} - \frac{\partial z_2'}{\partial \theta_{ij}'} \right) = -\left( \frac{\partial z_1}{\partial \theta_{ij}} - \frac{\partial z_2}{\partial \theta_{ij}} \right). \tag{37}$$

So we show that

$$\left[ \frac{\partial \mathcal{L}_{\text{align}}^{\text{BYOL}'}}{\partial \theta_{ij}'} \right]_{\parallel} = \left[ \frac{\partial \mathcal{L}_{\text{align}}^{\text{RAFT}}}{\partial \theta_{ij}} \right]_{\parallel}. \tag{38}$$

We differentiate $\mathcal{L}_{\text{cross-model}}^{\text{BYOL}'}$, $\mathcal{L}_{\text{cross-model}}^{\text{RAFT}}$ with respect to $\theta_{ij}'$, $\theta_{ij}$ respectively, we obtain that

$$\left[ \frac{\partial \mathcal{L}_{\text{cross-model}}^{\text{BYOL}'}}{\partial \theta_{ij}'} \right]_{\parallel} = -2 \left( \overline{z_2} - z_2' \right)_{\parallel} \cdot W' \cdot \frac{\partial f_{\theta'}(x_2)}{\partial \theta_{ij}'}, \tag{39}$$

$$\left[ \frac{\partial \mathcal{L}_{\text{cross-model}}^{\text{RAFT}}}{\partial \theta_{ij}} \right]_{\parallel} = 2 \left( \overline{z_2} - z_2 \right)_{\parallel} \cdot W \cdot \frac{\partial f_{\theta}(x_2)}{\partial \theta_{ij}}. \tag{40}$$

Similar to Eq. 35, we derive that

$$\left( \overline{z_2} - z_2' \right)_{\parallel} = \left( \overline{z_2} - z_2 \right)_{\parallel} \tag{41}$$

Since $\theta' = \theta$, $\partial f_{\theta'}(x_2)/\partial \theta'_{ij} = \partial f_{\theta}(x_2)/\partial \theta_{ij}$ and $W' = -W$, we have that

$$\left[ \frac{\partial \mathcal{L}_{\text{cross-model}}^{\text{BYOL}'}}{\partial \theta_{ij}'} \right]_{\parallel} = \left[ \frac{\partial \mathcal{L}_{\text{cross-model}}^{\text{RAFT}}}{\partial \theta_{ij}} \right]_{\parallel}. \tag{42}$$

Therefore by Eq. 38 and Eq. 42, RAFT's updating of the parameter $\theta$ is equal to BYOL$'$:

$$\left[ \frac{\partial \mathcal{L}_{\text{BYOL}'}}{\partial \theta'} \right]_{\parallel} = \left[ \frac{\partial \mathcal{L}_{\text{RAFT}}}{\partial \theta} \right]_{\parallel}. \tag{43}$$

Also we differentiate $\mathcal{L}_{\text{align}}^{\text{BYOL}'}$, $\mathcal{L}_{\text{align}}^{\text{RAFT}}$ with respect to $W_{ij}'$, $W_{ij}$ respectively, we obtain that

$$\left[ \frac{\partial \mathcal{L}_{\text{align}}^{\text{BYOL}'}}{\partial W_{ij}'} \right]_{\parallel} = 2 \left( z_1' - z_2' \right)_{\parallel} \cdot \left( \frac{\partial z_1'}{\partial W_{ij}'} - \frac{\partial z_2'}{\partial W_{ij}'} \right), \tag{44}$$

$$\left[ \frac{\partial \mathcal{L}_{\text{align}}^{\text{RAFT}}}{\partial W_{ij}} \right]_{\parallel} = 2 \left( z_1 - z_2 \right)_{\parallel} \cdot \left( \frac{\partial z_1}{\partial W_{ij}} - \frac{\partial z_2}{\partial W_{ij}} \right). \tag{45}$$

Note that $z_1 = -z_1'$, $z_2 = -z_2'$ and similar to Eq. 35, easy to show that

$$(z_1' - z_2')_\parallel = -(z_1 - z_2)_\parallel. \tag{46}$$

Also,

$$\frac{\partial z_1'}{\partial W_{ij}'} = \frac{\partial z_1}{\partial W_{ij}}, \tag{47}$$

$$\frac{\partial z_2'}{\partial W_{ij}'} = \frac{\partial z_2}{\partial W_{ij}}. \tag{48}$$

So we have

$$\left[\frac{\partial \mathcal{L}_{\text{align}}^{\text{BYOL}'}}{\partial W_{ij}'}\right]_\parallel = -\left[\frac{\partial \mathcal{L}_{\text{align}}^{\text{RAFT}}}{\partial W_{ij}}\right]_\parallel. \tag{49}$$

Differentiate $\mathcal{L}_{\text{cross-model}}^{\text{BYOL}'}$, $\mathcal{L}_{\text{cross-model}}^{\text{RAFT}}$ with respect to $W_{ij}'$, $W_{ij}$ respectively, we obtain that

$$\left[\frac{\partial \mathcal{L}_{\text{cross-model}}^{\text{BYOL}'}}{\partial W_{ij}'}\right]_\parallel = -2\left(\overline{z_2} - z_2'\right)_\parallel \cdot \frac{\partial z_2'}{\partial W_{ij}'}, \tag{50}$$

$$\left[\frac{\partial \mathcal{L}_{\text{cross-model}}^{\text{RAFT}}}{\partial W_{ij}}\right]_\parallel = 2\left(\overline{z_2} - z_2\right)_\parallel \cdot \frac{\partial z_2}{\partial W_{ij}}. \tag{51}$$

Then we have that

$$\left[\frac{\partial \mathcal{L}_{\text{cross-model}}^{\text{BYOL}'}}{\partial W_{ij}'}\right]_\parallel = -\left[\frac{\partial \mathcal{L}_{\text{cross-model}}^{\text{RAFT}}}{\partial W_{ij}}\right]_\parallel. \tag{52}$$

By Eq. 49 and Eq. 52, we prove that the cross-model loss of BYOL$'$ generates the opposite gradient to RAFT, namely,

$$\left[\frac{\partial \mathcal{L}_{\text{BYOL}'}}{\partial W'}\right]_\parallel = -\left[\frac{\partial \mathcal{L}_{\text{RAFT}}}{\partial W}\right]_\parallel. \tag{53}$$

Therefore by the two main conclusions Eq. 43 and Eq. 53, for BYOL$'$ with parameters $(\theta', W') = (\theta^{(0)}, W^{(0)})$ and RAFT with parameters $(\theta, W) = (\theta^{(0)}, -W^{(0)})$ respectively, we have

$$\text{BYOL:} \quad \left(\theta'^{(1)} = \theta'^{(0)} - \eta\left[\frac{\partial \mathcal{L}_{\text{BYOL}'}}{\partial \theta'}\right]_\parallel\bigg|_{\theta'=\theta'^{(0)}}, W'^{(1)} = W'^{(0)} - \eta\left[\frac{\partial \mathcal{L}_{\text{BYOL}'}}{\partial W'}\right]_\parallel\bigg|_{W'=W'^{(0)}}\right), \tag{54}$$

$$\text{RAFT:} \quad \left(\theta^{(1)} = \theta^{(0)} - \eta\left[\frac{\partial \mathcal{L}_{\text{RAFT}}}{\partial \theta}\right]_\parallel\bigg|_{\theta=\theta^{(0)}}, W^{(1)} = W^{(0)} - \eta\left[\frac{\partial \mathcal{L}_{\text{RAFT}}}{\partial W}\right]_\parallel\bigg|_{W=W^{(0)}}\right). \tag{55}$$

We derive that $\theta^{(1)} = \theta'^{(1)}$, $W^{(1)} = -W'^{(1)}$, and furthermore, $\theta^{(k)} = \theta'^{(k)}$, $W^{(k)} = -W'^{(k)}$ at any iteration $k$. In this way, we establish an one-to-one correspondence between the parameter trajectories of BYOL$'$ and RAFT in training, referred to as $\mathcal{H}$:

$$\mathcal{H} : \text{RAFT}_{(\theta, W)} \mapsto \text{BYOL}'_{(\theta, -W)} \tag{56}$$

# I    NON-TRIVIAL SOLUTIONS CREATED BY PREDICTOR

Suppose inputs $x_1 = t_1(x)$ and $x_2 = t_2(x)$ is $n$-dimensional. And in linear model, $f_\theta$, $f_\xi$ and $q_w$ is parameterized by matrices $(\theta_{ij})_{n \times n}$, $(\xi_{ij})_{n \times n}$ and $(W_{ij})_{m \times m}$ respectively.

The objective is

$$\mathcal{L}_{\text{BYOL}} = \mathbb{E}_{(x,t_1,t_2) \sim (\mathcal{X}, \mathcal{T}_1, \mathcal{T}_2)} \left[ \left\| q_w(f_\theta(t_1(x))) - f_\xi(t_2(x)) \right\|_2^2 \right]$$
$$= \mathbb{E}_{(x,t_1,t_2) \sim (\mathcal{X}, \mathcal{T}_1, \mathcal{T}_2)} \left[ \| W\theta x_1 - \xi x_2 \|_2^2 \right] \tag{57}$$

Differentiate $\| W\theta x_1 - \xi x_2 \|_2^2$ with respect to $\theta_{ij}$ and $W_{ij}$, we have

$$\frac{\partial \left[ \| W\theta x_1 - \xi x_2 \|_2^2 \right]}{\partial \theta_{ij}} = \sum_{k=1}^{m} \frac{\partial [W_{k,:}(\theta x_1) - \xi_{k,:} x_2]^2}{\partial \theta_{ij}}$$
$$= \sum_{k=1}^{m} 2 [W_{k,:}(\theta x_1) - \xi_{k,:} x_2] \frac{\partial [W_{k,:}(\theta x_1) - \xi_{k,:} x_2]}{\theta_{ij}}$$
$$= \sum_{k=1}^{m} 2 T_k W_{k,:}(x_1)_j$$
$$= 2 \left[ \left( W^\top T \right) x_1^\top \right]_{ij}, \tag{58}$$

$$\frac{\partial \left[ \| W\theta x_1 - \xi x_2 \|_2^2 \right]}{\partial W_{ij}} = \sum_{k=1}^{m} \frac{\partial [W_{k,:}(\theta x_1) - \xi_{k,:} x_2]^2}{\partial W_{ij}}$$
$$= \sum_{k=1}^{m} 2 [W_{k,:}(\theta x_1) - \xi_{k,:} x_2] \frac{\partial [W_{k,:}(\theta x_1) - \xi_{k,:} x_2]}{\partial W_{ij}}$$
$$= \sum_{k=1}^{m} 2 T_k (\theta x_1)_j \mathbf{1}_{\{k=i\}}$$
$$= 2 T_i \theta_{j,:} x_1$$
$$= 2 \left[ T(\theta x_1)^\top \right]_{ij}, \tag{59}$$

where $T_k = [W_{k,:}(\theta x_1) - \xi_{k,:} x_2]$, $T = (T_1, T_2, \ldots, T_m)^\top = W(\theta x_1) - \xi x_2$, and $W_{k,:}, \xi_{k,:}$ are the $k-$th row of $W$ and $\xi$ respectively.

So

$$\frac{\partial \left[ \| W\theta x_1 - \xi x_2 \|_2^2 \right]}{\partial \theta} = 2 \left[ \left( W^\top T \right) x^\top \right], \frac{\partial \left[ \| W\theta x_1 - \xi x_2 \|_2^2 \right]}{\partial W} = 2 \left[ T(\theta x)^\top \right] \tag{60}$$

Let

$$\frac{\partial \mathcal{L}_{\text{BYOL}}}{\partial \theta} = 0, \frac{\partial \mathcal{L}_{\text{BYOL}}}{\partial W} = 0, \tag{61}$$

we have that

$$\frac{\partial \mathbb{E}_{(x,t_1,t_2) \sim (\mathcal{X}, \mathcal{T}_1, \mathcal{T}_2)} \left[ \| W\theta x_1 - \xi x_2 \|_2^2 \right]}{\partial \theta} = \mathbb{E}_{(x,t_1,t_2) \sim (\mathcal{X}, \mathcal{T}_1, \mathcal{T}_2)} \left[ \frac{\partial \| W\theta x_1 - \xi x_2 \|_2^2}{\partial \theta} \right] = 0 \tag{62}$$

$$\frac{\partial \mathbb{E}_{(x,t_1,t_2) \sim (\mathcal{X}, \mathcal{T}_1, \mathcal{T}_2)} \left[ \| W\theta x_1 - \xi x_2 \|_2^2 \right]}{\partial W} = \mathbb{E}_{(x,t_1,t_2) \sim (\mathcal{X}, \mathcal{T}_1, \mathcal{T}_2)} \left[ \frac{\partial \| W\theta x_1 - \xi x_2 \|_2^2}{\partial W} \right] = 0 \tag{63}$$

When the weight of target $\xi$ converge, we have $\xi^{(k)} = \xi^{(k+1)}$ in the updating rule,

$$\xi^{(k+1)} = \tau_k \xi^k + (1 - \tau_k)\theta^{(k)}$$
$$\theta^{(k)} = \xi^{(k+1)} = \xi^{(k)} \tag{64}$$

Substituting $\xi$ by $\theta$, we obtain

$$\mathbb{E}_{(x,t_1,t_2)\sim(\mathcal{X},\mathcal{T}_1,\mathcal{T}_2)}\left[W^\top(W\theta x_1 - \theta x_2)x_1^\top\right] = 0 \tag{65}$$

$$\mathbb{E}_{(x,t_1,t_2)\sim(\mathcal{X},\mathcal{T}_1,\mathcal{T}_2)}\left[(W\theta x_1 - \theta x_2)\, x_1^\top \theta^\top\right] = 0 \tag{66}$$

Let $\mathbb{E}_{(x,t_1,t_2)\sim(\mathcal{X},\mathcal{T}_1,\mathcal{T}_2)}(x_1 x_1^\top) = A$, $\mathbb{E}_{(x,t_1,t_2)\sim(\mathcal{X},\mathcal{T}_1,\mathcal{T}_2)}(x_2 x_1^\top) = B$, we have that

$$eq: sylW^\top(W\theta A) = W^\top \theta B$$
$$W\theta A\theta^\top = \theta B\theta^\top$$
$$\Rightarrow W\theta - \theta BA^{-1} = \mathbf{0} \tag{67}$$

To solve Eq. **??** (which is called Sylvester's equation), we using the Kronecker product notation and the vectorization operator vec, we can rewrite the equation in the form

$$\left(I_m \otimes W - (BA^{-1})^T \otimes I_n\right) \operatorname{vec}\theta = \operatorname{vec}\mathbf{0} \tag{68}$$

So it has a non-trivial solution $\theta$ if and only if $\left(I_m \otimes W - (BA^{-1})^T \otimes I_n\right)$ has a non-trivial null space. An equivalent condition to having a non-trivial null space is having zero as an eigenvalue. Let $W$ has eigenvalues in common with $BA^{-1}$, then we have a non-trivial solution of $\theta$, which is exactly the prevention for collapse.

