# OpenReview forum: "Run Away From your Teacher: a New Self-Supervised Approach Solving the Puzzle of BYOL"
_ICLR.cc/2021/Conference — Reject_

### Official Review · AnonReviewer2 · 2020-10-23
**Review2**

**Rating:** 5
**Confidence:** 4

**Review:**

This paper mainly proposes an objective that incorporates the target network in BYOL in a opposite way that encourages the prediction of the online network to be far away from the target network.

Concerns:

1.  It is overly claimed that "we unravel the puzzle of how BYOL avoids representation collapse". For example, the authors fail to capture some intrinsic properties of BYOL, such as the role of prediction head (MLP + BN).  The theorem 5.1 can only deal with the prediction head that is linear. The authors could refer to the [1] for more insights.

2. The experimental results do not support the claim that "RAFT is a conceptually non-collapsing algorithm". In table 1, for results equipped with q_{w}-MLP, RAPT with better acc (71.31) in fact does not have smaller uniform loss. Instead, its alignment loss is smaller (which brings the acc improvement). So, the RAFT does not actually always enlarge the uniformity as claimed.

3. The implementation of BYOL in this paper regarding Cifar10 is not convincing. [2] also use the resnet18 as the encoder and it achieves the accuracy with 91+ in Cifar10. However, the reproduced result in this paper is only around 70.

4. BYOL proves its own effectiveness in ImageNet. To make fair comparisons, the authors shall conduct experiments in the same dataset. Otherwise, the claim regarding to the BYOL might not be solid.

5. It is acceptable that running away from the mean teacher increases the difficulty of alignment. But it is unclear to the reviewer why it can produce global uniformity in the representation space?

[1] Tian, Yuandong, et al. "Understanding Self-supervised Learning with Dual Deep Networks." arXiv preprint arXiv:2010.00578 (2020).

[2] Ermolov, Aleksandr, et al. "Whitening for Self-Supervised Representation Learning." arXiv preprint arXiv:2007.06346 (2020).

---

> ### Author Response · Authors · 2020-11-21
> **Reply to R2**
>
> We would like to thank R2 for the valuable questions and advice. We would like to respond to your concerns one by one in this post.
>
> ## Concern 1
> Thanks for pointing out that our paper’s title oversells the contributions we made in our work. We sincerely apologize for that. While besides the title itself, we are meticulous in the statement we make in the main text of our paper: we avoid  claiming we solve the problem why BYOL works, and we further point out that RAFT is just “conceptually working”, but its success is also dependent on the predictor. As for the reason why our main focus is on the predictor instead of other elements, we would like to give an informal explanation.
>
> **Why we don’t analyze BN in the predictor:** the blog post[1] which motivates the author of [2] states in its main content clearly shows that when the BN is removed from the predictor, BYOL continues to work (refer to Table “Performance for each variation”).
>
> **Why we don’t analyze BN in general:** firstly according to the appendix of the blog post[1], when the BN is both removed from the predictor and the projector, training for longer epochs would make the model recover. Secondly, the author of [2] also states in the openreview reply that the BN is just a “sufficient condition”, which might not be crucial to the success of BYOL. Besides, the original author of BYOL also published another version of BYOL that doesn’t require the batch statistics at all[3], indicating that BN might not be the most essential component.
>
> ## Concern 2
> We don’t focus on the relative performance between three algorithms we evaluate in the paper (BYOL, BYOL’, and RAFT). However, the only comparison that’s crucial to our conclusion is the comparison between the algorithms and the random baseline. We focus on whether the algorithm works or not, e.g. whether the representation collapse happens. RAFT is a more conceptually working algorithm lies in one analysis and one experimental result:
>
> **Analysis:** In Figure 2b (2a in the latest version), when two samples are mapped to have initial different gradient, the RAFT would separate them in the next couple of iterations.
>
> **Experimental Result:** In Table 1, the RAFT loss remains to be effective regularizing the alignment loss, which indicates that RAFT is more unified learning objective compared to BYOL and BYOL’.
>
> ## Concern 3
> Due to the limitation of our computational resources, we don’t train our method as [4] in 1000 epochs. Instead we train them in 300 epochs, which could be told by the trend of the accuracy doesn’t achieve the best performance since BYOL performs better when trained longer. More importantly as we stated in the response to Concern 2, we only care whether the algorithm outperforms the random baseline, since that’s our main focus on explaining how BYOL avoids the representation collapse.
>
> ## Concern 4
> Thank you for pointing out that evaluating the algorithms by other large-scale datasets would make our work more solid. We are trying to gather more computational resources to evaluate our proposed RAFT and BYOL. However we would like to also point out that based on the experimental results of BYOL paper and our paper, the dataset would not change BYOL from collapse to non-collapse, nor in the other way. Thus in this respect, the dataset is not so crucial to our final conclusion. Again, we thank the reviewer for this advice and we would improve the solidness of our work in the near future.
>
> ## Concern 5
> Thank you for noticing this phenomenon (which shows that you understand our work from the alignment-uniformity framework, and that’s what really makes us happy)! Unfortunately, this is also unclear to us. Our paper disentangles the analysis of BYOL into two separate parts:
> - When is BYOL approximately equivalent to BYOL’?
> - How does the predictor help produce the representation uniformity?
> While these two problems are also challenging, our main contribution is that we provide empirical results supporting the legitimacy of these two parts. We don’t expect that we completely solve them in a single paper, which means we leave the job of analyzing them to the future.
>
>
> [1] Abe Fetterman & Josh Albrecht. “Understanding self-supervised and contrastive learning with "Bootstrap Your Own Latent" (BYOL).” https://untitled-ai.github.io/understanding-self-supervised-contrastive-learning.html
> [2] Tian, Yuandong, et al. "Understanding Self-supervised Learning with Dual Deep Networks." arXiv preprint arXiv:2010.00578 (2020).
> [3] Pierre H. Richemond et al. “BYOL works even without batch statistics.” arXiv preprint arXiv:2010.10241
> [4] Ermolov, Aleksandr, et al. "Whitening for Self-Supervised Representation Learning." arXiv preprint arXiv:2007.06346 (2020).

---

### Official Review · AnonReviewer3 · 2020-10-26
**RAFT**

**Rating:** 3
**Confidence:** 5

**Review:**

This paper analyses the recently proposed Bootstrap Your Own Latent (BYOL) algorithm for self-supervised learning and image representation.
The authors first derive an alternative training procedure called BYOL' by computing an upper bound of the BYOL objective function.
After diverse analyses, the authors then introduce Run Away From Your Teacher (RAFT), where RAFT is another BYOL variant that resembles contrastive method by having an attractive and repealing term in the training objective. According to the authors, this decomposition allows for a better understanding of the training dynamics.

Finally, the authors made the following transitivity reasoning:
 - BYOL and BYOL' are almost equivalent
 - RAFT and BYOL' are shown to be equivalent under some assumptions.
Thus, conclusions that are drawn from analyzing RAFT should still hold while analyzing BYOL. They thus link the interest of BYOL's predictor and the EMA through the RAFT loss decomposition.

I have multiple strong concerns regarding this paper. These concerns are both on the paper results, shortcuts in the analysis, and the writing style.


Results:
--------------

 - In section 4, the authors introduce BYOL' as a variant of BYOL. To do so, they derive an upper bound on the BYOL loss, i.e. the L2 distance between the projection and the projector, and they try to minimize it. However, this approach disregards that BYOL does not minimize a loss (due to the stop gradient). In other words, the BYOL objective keeps evolving during training; the target distribution is non-stationary.  As mentioned in the BYOL paper: "Similar to GANs, where there is no loss that is jointly minimized w.r.t. both the discriminator and generator parameters; there is therefore no a priori reason why BYOL’s parameters would
converge to a minimum of L_BYOL given the online and target parameters". Minimizing an upper-bound is at best insufficient, at worst a non-sense. The sentences, "minimizing L_{BYOL'}  would yield similar performance as minimizing L_{BYOL}" and "we conclude that optimizing L_{BYOL'} is almost equivalent to L_{BYOL}" are unfortunately wrong.  This is somewhat highlighted different qualitative results in Appendix F.1.b != F.1.d.
A better approach would be to ensure that the *gradients* go in a similar direction (so the training dynamics are similar rather than the objective function). However, even such a demonstration could be insufficient due to compounding factors in the training dynamics.
 - The 1-1 mapping between BYOL' and RAFT rely on three hypotheses. While (i) and (ii) are reasonable, hypothesis (iii) is quite strong, and more importantly, neither elaborated nor discussed. In other words, I am unable to validate/invalidate the interest of the theoretical results. Would it be possible to measure the normal gradient empirically? To bound it?
 - In section 3, i would recommend the author to mention that multiple components were also in the BYOL paper; especially when writing "therefore, we conclude the predictor is essential to the collapse prevention of BYOL."
 - Although I acknowledge that self-supervised learning requires heavy computational requirement, and few teams may run experiments on ImageNet. Yet, I would recommend the authors to not use CIFAR10 as the dataset has multiple known issues (few classes, small images, few discriminative features). Other variants such at STL or ImageNete can be trained on a single GPU over a day, and are less prone to misinterpretation in the results. Besides, I want to point out that BYOL was not correctly tuned: the experiments are based on a different optimizer (Adam vs LARS) and no cosine decay were used for the EMA, while these two components seem to be critical, as mentioned in BYOL and arxiv:2010.1024.

Overall, I have a serious concern about the paper's core contributions. However, there are still some good elements in the paper that I think are under-exploited:
 - RAFT is itself an original, new and interesting algorithm. The potential link to BYOL is indeed an interesting lead, but in its current state, I would make it a discussion more than a key contribution.
 - Table D.3 shows that RAFT/BYOL' does not collapse without predictors when \beta is high. Albeit providing low accuracy, a non-collapse is quite surprising. Unfortunately, the authors leave it for future work


Shortcuts:
--------------
I was surprised by multiple shortcuts in the reasoning process or undiscussed conclusions:
 - The authors mention that the predictor is a dissatisfactory property of BYOL. Could they elaborate? This is actual the key component of the method (if not the only one!), and such pro/cons could be detailed in light of other methods.
 - In section 4.1, the authors mention that: similar accuracies and losses are sufficient somewhat confirm that BYOL and BYOL' are similar. Two completely different methods may have the same errors while being radically different...
 - In Section 4.2, the authors mention that "Based on the form of BYOL, we conclude that MT is used to regularize the alignment loss". However, there is no experiments to try to contradict/validate this claim. Differently, the EMA may ease the optimization process or it may have different properties. Even if I understand the logic behind this statement, I regret that the authors do not try to confort it.
- In section 4.2, the authors mention that there exist multiple works (while only citing one...) demonstrating that EMA is "roughly" equivalent to sample averaging and may encourage diversity. While this is sometimes true in specific settings (cf. markov game and fictitious play), this is also known to ease optimization (cf. target network in DQN). Stating that RAFT is better than BYOL because it better leverage the EMA target is tricky without proper analysis.
- Albeit understandable, the transitivity between BYOL and RAFT is difficult to defend due to multiple approximations and hypothesis. Therefore, it is of paramount importance that the approximations and hypothesis are validated, which is not sufficiently done in the paper.


Writing:
--------------
 - Although papers' writing quality remain subjective, I tend to expect a formal language. I kind of feel ill-at-ease when reading sentences including "BYOL works like a charm", "disclosing the mistery", "to go harsher", "bizarre phenomon". Other sentences also expresses judgement such as "inconsistent behavior", "dissatisfactory property of BYOL" or "has admirable property" without proper argumentation.
 - It is non-trivial to follow the different version of the algorithms... which are defined in the appendix. Please consider renaming BYOL'.
 - A related work section would have been useful to put in perspective BYOL that are theoretically motivated e.g. AMDIM, InfoMin, other self-supervised learning methods without negative example, e.g. DeepCluster, SwAV. Section 2 is more about the background, not related work.
 - there are a few confusions in the notation, \alpha \beta have different meaning across equations (Eq 7 vs 8)
 - In section 3, random is ill-defined. In Cifar10, random should be 10%, I assume that you refer to random projection. Please clarify.
 - Figure 1 is clear, and I recommend to keep it as it is.
 - From my perspective, the mathematical explanation in Section 5 is quite obfuscated, and I would recommend a full rewriting.
 - Please avoid unnecessary taxonomy, e.g. uniformity optimizer, effective regulizers and others.
 - In conclusion, you mentioned some results about the projector. However, you never detail them in the paper. Please, do not discuss unpublished results.

Overall, I had difficulties following the paper: I keep alternating between the appendix, previous sections, and the text. Again, the phrasing makes me ill-at-ease.


Conclusion:
--------------------
I have some serious concerns about the core results of the paper. Importantly, Theorem 4.1 follows a misinterpretation of the BYOL training dynamics. From my perspective, there are too many unjustified claims, and I cannot recommend paper acceptance. However, there is some good idea in the paper, and I strongly encourage the authors to study RAFT independently of BYOL in the future.

---

> ### Author Response · Authors · 2020-11-21
> **Reply to R3 (part 1/3)**
>
> Dear R3, we appreciate the patience of you reading our paper and giving such detailed feedback. There is much helpful advice and many constructive ideas in your review. **However, we feel sorry that your judgement of our work may be conditioned on the misunderstanding of the evaluation metric “linear evaluation protocol” in the self-supervised learning field (and other reviewers seem not having the similar concern).** We would like to address your concerns in our following reply.
>
> ## Linear Evaluation Protocol and Random Baseline
> We are somewhat concerned about the fact that the reviewer doesn’t understand the widely used  evaluation metric in self-supervised learning, which is reflected by his/her misunderstanding of “random baseline”. In the “Writing” part, the reviewer writes,
> > In section 3, random is ill-defined. In CIFAR10, random should be 10%, I assume that you refer to random projection. Please clarify.
>
> Here we would like to clarify why the “random baseline” is not 10%.
>
> Most of work in the field of self-supervised learning adopts the evaluation metric called “linear evaluation protocol” to estimate the quality of the representation distribution. Normally, after training under the pretext task, we would yield an encoder network (or feature extractor). To evaluate how well this encoder network is, we fix the weights of it, and then add a linear layer on top of the encoder to train a classifier on the labeled dataset. The point of the linear evaluation protocol is to see whether the data of the same class can be mapped to the representation space so that they could be easily identified. Therefore if we randomly initialize a network and evaluate it under the linear evaluation protocol, we would normally yield better classification accuracy than the RANDOM CLASSIFIER (which has 10% of accuracy on CIFAR10), due to the natural pixel-level intra-class similarity. In our paper, we clearly stated the setting at the beginning of Section 3:
> > The performance of BYOL original model, whose predictor $q_w$ is a two-layer MLP with batch normalization, evaluated on the linear evaluation protocol (Kolesnikov et al., 2019; Kornblith et al., 2019; Chen et al., 2020a; He et al., 2020; Grill et al., 2020) reaches 68.08 ± 0.84%. When the predictor is removed, the performance degenerates to 20.92 ± 1.29%, which is even lower than the random baseline’s 42.74 ± 0.41%.
>
> Some may argue that the misunderstanding of the evaluation metric is caused by our poor writing, while in the section 3 of the original paper of BYOL[1], the author’s description is of the same style of ours, which can’t be simpler and clearer:
> > We apply this procedure by predicting a fixed randomly initialized network achieves 18.8% top-1 accuracy (Table 5a) on the linear evaluation protocol on ImageNet, whereas the randomly initialized network only achieves 1.4% by itself.
>
> According to the reviewer, since the ImageNet is of 1000-class classification task, then the random baseline should have 0.1% of accuracy instead of 1.4%, while in reality it's not the case.
>
> The basic understanding of the evaluation metric is fundamental to fair evaluation. In our case, misunderstanding the linear evaluation protocol would lead to misunderstanding the concept of the representation collapse, which is more crucial to rating our contributions. And this consequential misunderstanding is also reflected in the reviewer No.3’s response.
> In the “Results” part, the reviewer writes,
> > Table D.3 shows that RAFT/BYOL' does not collapse without predictors when $\beta$ is high. Albeit providing low accuracy, a non-collapse is quite surprising. Unfortunately, the authors leave it for future work.
>
> In fact, in our paper, we introduce the alignment-uniformity framework[2] to readers to understand the concept of representation collapse. The representation collapse means that most of the data are mapped to the same meaningless point, which can be reflected by the metric of uniformity. When the predictor is removed, BYOL, BYOL’ and RAFT are all collapsed since their uniformity loss is much higher than the random baseline: -0.14, -0.10, and -0.006 respectively, while the random baseline even has -0.51 of uniformity.
>
> We believe that the basic understanding of the linear evaluation protocol and the representation collapse is crucial to the objectiveness of the review. And therefore we sincerely hope that the reviewer could re-evaluate our work after reading our reply.
>
> [1] Jean-Bastien Grill, Florian Strub, Florent Altche ́, Corentin Tallec, Pierre H Richemond, Elena Buchatskaya, Carl Doersch, Bernardo Avila Pires, Zhaohan Daniel Guo, Mohammad Gheshlaghi Azar, et al. Bootstrap your own latent: A new approach to self-supervised learning. arXiv preprint arXiv:2006.07733, 2020.
>
> [2] Tongzhou Wang and Phillip Isola. Understanding contrastive representation learning through alignment and uniformity on the hypersphere. arXiv preprint arXiv:2005.10242, 2020.

---

> ### Author Response · Authors · 2020-11-21
> **Reply to R3 (part 2/3)**
>
> ## Response to the Valuable Advice & Other Comments
> We would like to address the concerns raised by R3 in the “Results” and “Shortcuts” section. As for the “Writing”, we apologize if our phrasing somehow disturbs you. We will try to mild our excitement of finding that optimizing two opposite losses would yield the same effect and shift to a more formal language. Below we list the summarization and the response, please correct us if we misunderstand your advice.
>
> ### Result (1)
> **Review.** The reviewer summarizes the BYOL’ upper bound loss as the distance between the “projection and the projector”. The reviewer thinks that BYOL itself doesn’t minimize a loss due to the stop gradient, which the reviewer thinks is supported by the non-stationary distribution of the target network and the non-convergence observed in BYOL. Therefore, the reviewer concludes that analyzing and altering the loss would be wrong, which is supported by our visualization results in appendix F.1 and offers us a better direction of analyzing BYOL: gradient.
>
> **Reply.** Firstly, the reviewer wrongly summarizes the BYOL’ upper bound loss. In fact, the BYOL’ loss is composed of two objectives: (i) attracting the representations of the two views of the same data after the predictor and (ii) repealing the online and the MT under the same data distribution.
>
> Secondly, it’s hard to understand the point of the reviewer’s claim that “BYOL doesn’t minimize a loss”. What’s more, the reviewer’s theory “since the target network is non-stationary, BYOL doesn’t minimize a loss” is just stating a conditional phenomenon observed by the researcher, but is a completely wrong causal inference: if we remove the predictor from BYOL, the target network remains “non-stationary”, but BYOL indeed minimizes a loss: the loss of BYOL quickly goes to zero and follows the representation collapse (refer to our result in appendix). Our way of analyzing BYOL starts from observing the two quantifiable metrics that have been shown crucial to the contrastive-based methods: alignment and uniformity. Though BYOL does not explicitly optimize them, we find that these two metrics are indeed optimized during training by estimating them. The whole point of analyzing BYOL lies in how we can relate its training objective to the alignment-uniformity framework, and approximately equating BYOL and BYOL’ is empirically supported, but not theoretically. We would like to discuss under what condition this approximation holds in the future, while in this paper our main focus is to provide an overall understanding framework for BYOL.
>
> Thirdly, we would like to point out, negating the claim that BYOL is approximately equivalent to BYOL’ by the qualitative results presented in F.2 is itself a shortcut: retraining the neural network on the same dataset with different random seed would yield the close performance while probably different qualitative results. The reason why we present the qualitative results in the appendix is that we would like to show the apparent difference between the collapsed methods and the working one.
>
> At last, we agree that analyzing BYOL from the perspective of the gradient would be a good direction, while our approximating and redesigning the loss function would consequently cause effects on the gradient. There is no contradiction between these two methods. We would like to investigate the problem with your advice in the near future.
>
> ### Result (2)
> Since the representation space is a hypersphere, the concentration and separation of the data samples are mainly influenced by the tangential component of the gradient. This condition is trivial and easy to satisfy. Suppose the unit vector $z$ is the representation produced by the online network and the unit vector $\bar{z}$ is the representation produced by the MT. The original loss is:
>
> \begin{align}
> \mathcal L = \big|\big| z - \bar{z} \big|\big|_2^2
> \end{align}
>
> After applying the condition using simple mathematical techniques, the loss is changed to:
>
> \begin{align}
> \mathcal L = \frac{1}{\langle z, \bar{z}\rangle} \big|\big| z\langle z,\bar{z}\rangle - \bar{z} \big|\big|_2^2
> \end{align}
>
> Where the inner-product $\langle z, \bar{z} \rangle$ is a scalar and doesn’t generate any gradient. Our additional experiments show that this condition also doesn’t affect whether the algorithm would collapse. Please refer to our latest version of the paper.
>
> ### Result (3)
> Our claim “the predictor is essential to the collapse prevention of BYOL” is based on the observation that when the predictor is removed, the collapse happens. Other factors are also important to the final quality of the representation distribution, while they do not essentially affect whether the algorithm would collapse, which is also supported by the original BYOL paper Table.5b[1].

---

> ### Author Response · Authors · 2020-11-21
> **Reply to R3 (part 3/3)**
>
> ### Result (4)
> Firstly, thank you for the advice that adopting other datasets other than CIFAR10 would increase the reliability of our experimental results and we are aware of that. We are currently actively searching for more computational resources to validate the effectiveness of our RAFT algorithm, while we would like to point out that the data distribution (dataset) will not change whether a method collapses or not. If any, as mentioned by the reviewer, the poor quality of CIFAR10 (“few classes, small images, few discriminative features”) only increases the difficulty for an algorithm to work. Most importantly, being aware of the shortcoming of the dataset, we never claim the superiority of the RAFT in terms of its better capability generating good-quality representations. Instead, the RAFT loss is more unified from the perspective of algorithm designing: we know alignment and uniformity are two terms regularizing each other, and we want to leverage the EMA of the history models, would we choose to fit to it or run away from it? The answer is pretty clear.
>
> Secondly, the same logic applies when we address the concern you raised: “BYOL was not correctly tuned.” The optimizer doesn’t affect the property whether the model collapses or not, which is demonstrated by the BYOL original paper and our experimental results. The cosine decay trick is only crucial in terms of the final performance: with or without it, the model effectively works better than the random baseline. Therefore we don’t consider discussing them as we focus on the essential component making BYOL avoid collapse.
>
> ### Shortcut (1)
> We sincerely apologize if our phrasing causes your misunderstanding on our claim that "predictor is a dissatisfactory property of BYOL". When we say "dissatisfactory" we mean that the current explanation on BYOL doesn't consider the predictor seriously. By making this argument, we want to emphasize the importance of the predictor and what we did in this work was to at least show the efficacy of it with respect to the fact that the predictor helps establish the equivalence between BYOL' and RAFT. One main contribution of our work is to emphasize that there are some additional unexpected effects brought by this predictor, and the equivalence between BYOL’ and RAFT is one of them. Analyzing the efficacy of the predictor is a must if we want to fully understand how BYOL works.
>
> ### Shortcut (2)
> We acknowledge that the approximate equivalence between BYOL and BYOL’ is supported only by the empirical study. While given the similarity between the two losses and the upper bounding relation, the closeness between the two is somewhat obvious. We would like to explore under what condition this equivalence holds in the near future.
>
> ### Shortcut (3)
> In our paper, our claim is made on the BYOL’, which is composed of two loss terms, but not BYOL itself. Based on the previous work of alignment-uniformity framework, that the alignment is regularized by the uniformity objective, our claim that cross-model term regularizes the alignment term is self-contained.
>
> ### Shortcut (4)
> The RAFT loss is better than BYOL in terms of leveraging the MT because when the predictor is removed, running away from MT remains to be an “effective regularizer” for the alignment loss, thus is a more unified method compared to BYOL and BYOL’.
>
> ## About Writing
> Again, we are extremely sorry if our writing disturbs you. Our intention of writing this paper was to put all the materials in the main text. Unfortunately, due to the limitation of length, we have to put some of the empirical evidence and theoretical proof to the appendix. We will upload another version of our work to arxiv that properly addresses our writing problem. Besides, we only mention the word “projector” in the appendix, never in the conclusion. We believe the reviewer’s accusation of our discussing unpublished results is unintentional and we fully understand it.
>
> To summarize, we are grateful that R3 provides so many important reviews and questions. However, we sincerely request a full re-evaluation of our work after our clarification on the misunderstanding.

---

### Official Review · AnonReviewer4 · 2020-10-28
**Studying an interesting problem, however the main claim is erroneous**

**Rating:** 3
**Confidence:** 4

**Review:**

**Summary**: the paper aims to explain the success of BYOL, a recently proposed contrastive method that mysteriously avoids the trivial constant solution without requiring negative samples. The paper proposes a new loss named RAFT. Compared to BYOL, RAFT is more general since it subsumes a variation of BYOL as its special case, and contains a cross-model term to be maximized which regularizes the alignment loss and encourages the online encoder to "run away" from the mean teacher.

The paper claims this cross-model term encourages disparity, which could help demystify why BYOL does not collapse to a trivial solution. However, the cross-model term itself cannot prevent outputs from collapsing, as explained below.

**Question 1**: my main concern is the effectiveness of the cross-model loss: I disagree that the cross-model loss prevents collapsing representations. I think the authors may be confusing contrasting two samples ("cross-sample") and contrasting two functions ("cross-model"):
- The cross-model loss is essentially the L2 distance between two functions, which is the average squared error between two model outputs on the *same sample.*
- The common contrastive loss, which contrasts outputs from the same model on *different samples*.

For example, suppose MT is a constant function at the $t_{th}$ iteration (i.e. the function outputs some constant $c$ for all input), then the online encoder could be updated to be another constant as far away from $c$ as possible, i.e. the cross-model loss is maximized, however we still have the sample collapsing issue. As a side note, a constant function also achieves a perfect alignment loss.
More concretely, consider $f(x) = Wx +b$ where $W$ is initialized to be the all-0 matrix, i.e. $f(x) = b$ is a constant function. Then for all future updates, learning $f$ only updates $b$ but not $W$ (since there's no gradient on $W$), and therefore $f$ will remain a constant function, i.e. it always collapses the points. One may argue that it is wrong to choose $W = 0$, but the point is, the success of BYOL needs more careful analysis of the optimization process, which cannot be addressed by the cross-model loss term itself.

**Question 2**: section 3 phrases the need of a predictor as a disadvantage of BYOL, however RAFT also requires a predictor head to achieve good classification performance. Studying the effect of the predictor is an interesting direction and will make the paper much stronger, as the authors also point out in the conclusion.

**Other comments**:
- Table 3: why are there no results for BYOL'-MLP? Comparing RAFT-NP to BYOL'-NP, there doesn't seem to be a clear edge of RAFT, both in terms of the uniformity loss and the accuracy.
It would also be better to highlight the key results in the table. The current table is quite dense; adding more highlights and comments will have the reader understand what to take away from these results.
- Paper organization: a lot of material is deferred to the appendix, which makes the paper a bit hard to follow since the reader needs to jump back and forth. It would be better if results in the appendix are better summarized in the main text.
- The term BYOL' first appears at the end of the first paragraph on page 2 without a definition.
- Minor typo: there's an extra left parentheses in front of the second $f$ in equation (5); an extra comma after "distribution" in the first paragraph of section 3.

---

> ### Author Response · Authors · 2020-11-21
> **Reply to R4 (part 1/2)**
>
> Thank you for the valuable advice. It’s sad that you only noticed the weaknesses of our paper and ignored all the contributions we’ve made. We would like to address your major concerns through this reply and recapitulate our contribution.
>
> ## Question 1
>
> **Question.** The reviewer thinks we are unaware of the difference between contrasting two samples (“cross-sample”) and contrasting two functions (our “cross-model” loss). Then the reviewer argues that contrasting two functions is not capable of collapse prevention by giving a corner case where the weight of the matrix is initialized to zero. The point of this “zero-initialized” example is to emphasize that special care is required when dealing with the cross-model term.
>
> **Reply.** Firstly, the example given is baseless and overly critical. As known by every practitioner in deep learning, neural networks require a reasonable initialization scheme, and zero-initialization often completely disables a network. The reviewer’s reasoning is applicable to attacking any form of losses, even BYOL itself: if there is an intermediate layer whose weight is zero in the BYOL, then the loss would be a zero constant and nothing would be learned during training.
>
> We understand the reviewer’s concern and we are prudent with our conclusions. There are some achievable conditions required when we claim that maximizing the cross-model term could prevent collapse, and the randomness of the initial representation distribution is one of them. What’s more, we shall emphasize that maximizing the cross-model loss is not unconditionally equivalent to contrasting two samples in the representation space, and we avoid claiming it in the paper. We argue that only when the EMA is considered in the target network, the cross-model loss is able to contrast samples. In section 4.2, we leverage the conclusion in the Mean Teacher[1], which is also mentioned in R3, to bridge the gap between the sample averaging and model averaging:
> > There has been a lot of work demonstrating that weight averaging is roughly equal to sample averaging[1], thus if two samples’ representations are close to each other at the beginning and their initial updating directions are opposite, then RAFT consistently separates them in the representation space.
>
> [1] Antti Tarvainen and Harri Valpola. Mean teachers are better role models: Weight-averaged consistency targets improve semi-supervised deep learning results. In Advances in neural information processing systems, pp. 1195–1204, 2017.

---

> ### Author Response · Authors · 2020-11-21
> **Reply to R4 (part 2/2)**
>
> ## Question 2
>
> We sincerely apologize if our phrasing causes your misunderstanding on our claim that "predictor is a dissatisfactory property of BYOL". When we say "dissatisfactory" we mean that the current explanation on BYOL doesn't consider the predictor seriously. By making this argument, we want to emphasize the importance of the predictor and what we did in this work was to at least show the efficacy of it with respect to the fact that the predictor helps establish the equivalence between BYOL' and RAFT.
>
> We agree that RAFT doesn’t completely solve the problem of collapse when the predictor is removed. Our claim that RAFT is a more unified objective (refer to the Figure.2(b) in the latest version of our paper) and is thus more “essential” is based on the previous work of alignment-uniformity framework[2], which demonstrates that the alignment loss and the uniformity loss are two competing factors regularizing each other. Evaluate the role of RAFT and BYOL’ from the perspective of algorithm designing: suppose you want to use another term to constrain the alignment loss which incorporates the Mean Teacher, would you choose to fit to it or run away from it? RAFT loss remains to be an effective regularizer when the predictor is removed, while BYOL’ fails to do so, which implies that RAFT is more favorable.
>
> Yes, this conclusion still has potential to be improved by enabling the framework to work when the predictor is removed, while demanding our single paper to solve all the problems would be unfair, let alone under the 8-page limitation. Our further analysis on the efficacy of the predictor focuses on the equivalence between BYOL’ and RAFT, which makes two crucial contributions:
> The importance of the predictor lies in at least equating the two opposite training objectives. And we emphasize the efficacy of the predictor needs to be further studied.
>
> There are multiple factors entangled in BYOL. Our contribution provides a novel view to investigate it. Under our framework, the direction of analyzing BYOL is much clearer than before: two separate questions wait to be answered in the future:
> - Under what condition BYOL is equivalent to BYOL’?
> - How does the predictor help optimizing the uniformity of the representation in RAFT?
>
> We hope our explanation on the RAFT loss and our framework of understanding the working mechanism behind BYOL would provide a new direction of leveraging the MT in the future. We would also like to thank R4 for the two valuable comments, which we think would be helpful to our future study.
>
> [2] Tongzhou Wang and Phillip Isola. Understanding contrastive representation learning through alignment and uniformity on the hypersphere. arXiv preprint arXiv:2005.10242, 2020.

---

### Official Review · AnonReviewer1 · 2020-10-30
**Official Blind Review**

**Rating:** 6
**Confidence:** 3

**Review:**

*Summary*
The paper provides a new perspective on the BYOL self-supervised learning method. First, the paper introduces an upper-bound objective, BYOL', that is easier to analyze than BYOL because it is composed of two well understood losses: an alignment loss and cross-model loss. Further, it shows empirically that optimizing BYOL' is similar to optimizing BYOL. Second, the paper introduces the RAFT method which maximizes the alignment loss instead of minimizing it. The paper proves that under some assumptions, such as a linear predictor function, optimizing BYOL' is equivalent to RAFT. Based on this analysis, the paper explains why the predictor function is essential for BYOL and why it is hard to achieve convergence.

*Quality*
I really like the analysis of the paper. The paper provides a mix of theoretical and empirical argument for understanding BYOL, and introduces a new method called RAFT. The main drawback of the paper is that it limits the empirical analysis to a single and much simpler experimental setup using CIFAR10 and resnet18. I believe that since BYOL's significance is an empirical one and is mainly established on Imagenet, any empirical analysis of BYOL in other simpler settings is quite limited.

*Clarity*
The authors have done a very good job in writing this paper. The logic, presentation and results are quite clear to understand.

*Originality*
I find the paper quite interesting and original in its analysis. I especially like the analysing BYOL through the BYOL' upper-bound.

*Significance*
I think the results of the paper could have been quite more significant if applied on other experimental setups. While I understand working with SOTA models can be computationally expensive, the main argument of this line of work is empirical and it is hard to be convincing without more extensive empirical results.

---

> ### Author Response · Authors · 2020-11-21
> **Reply to R1**
>
> Dear R1,
>
> Thank you for the kind and constructive comments. We are also aware of your concern whether RAFT is comparable to the SOTA methods when applied to the large-scale datasets such as ImageNet. The main point of our paper, however, focuses on why BYOL doesn’t collapse. Larger datasets evaluate the effectiveness of the algorithm, while using smaller one is sufficient to demonstrate whether the algorithm would collapse or not: if our algorithm performs better than the random baseline and close to the BYOL, then we are confident to claim it’s a non-collapsing framework. On the other hand, if the algorithm fails (by fail, we mean worse than the random baseline) on CIFAR10, then even if it works on ImageNet, we would still regard it as flawed since it heavily depends on the data distribution. As for evaluating RAFT on ImageNet, we are trying to gather more computational resources to validate our proposed method, although we wouldn’t view RAFT’s effectiveness as the all-important contribution of this work.
>
> Being aware of the limitation brought by the dataset, we avoid claiming that RAFT is better than BYOL in terms of its performance on the linear evaluation protocol. The value of our work lies in the attempt of subsuming BYOL into the already-verified alignment-uniformity framework, and jumping out of the current understanding frameworks originally provided by BYOL under mild conditions, including Teacher-Student framework, DQN’s Online-Target framework and Mean Teacher in semi-supervised learning.

---

### Decision · Program_Chairs · 2021-01-07
**Final Decision**

**Decision:**

Reject

**Comment:**

Most of the reviewers and AC found many claims of this submission unsubstantiated.